# Image Captioners Are Scalable Vision Learners Too

**Michael Tschannen**[∘,†]    **Manoj Kumar**[∘]    **Andreas Steiner**[∘]
**Xiaohua Zhai**    **Neil Houlsby**    **Lucas Beyer**[∘]
Google DeepMind

## Abstract

Contrastive pretraining on image-text pairs from the web is one of the most popular large-scale pretraining strategies for vision backbones, especially in the context of large multimodal models. At the same time, image captioning on this type of data is commonly considered an inferior pretraining strategy. In this paper, we perform a fair comparison of these two pretraining strategies, carefully matching training data, compute, and model capacity. Using a standard encoder-decoder transformer, we find that captioning alone is surprisingly effective: on classification tasks, captioning produces vision encoders competitive with contrastively pretrained encoders, while surpassing them on vision & language tasks. We further analyze the effect of the model architecture and scale, as well as the pretraining data on the representation quality, and find that captioning exhibits the same or better scaling behavior along these axes. Overall our results show that plain image captioning is a more powerful pretraining strategy than was previously believed.

## 1   Introduction

Contrastive language image pretraining (CLIP) [50] has recently become a very popular pretraining strategy, enabling scalable vision-language pretraining on billions of image-text pairs collected from the web. CLIP not only enables zero-shot transfer to arbitrary image classification and retrieval tasks, but also produces high-quality vision backbones rivaling the best label-supervised ones [14]. The corresponding checkpoints [50, 9, 58] are among the most powerful publicly available vision backbones, enjoying wide adoption in the context of multi-modal vision-language models, where often a pretrained vision backbone is combined with a pretrained language model [1, 6, 62, 23, 17, 5].

Before contrastive image-text pretraining was popularized by CLIP [50] and ALIGN [26], a number of works explored generative image captioning [13, 54] and n-gram prediction [27, 37] approaches for representation learning at small scale. However, [50] showed that the zero-shot classification performance of contrastive pretraining scales much better than captioning as a function of the number of training examples seen (see [50, Fig. 2]). Since then, follow-up works of [50] which focus on pretraining from scratch do usually not rely on captioning alone, but augment it with a contrastive loss [66, 34, 40]. Those follow-up works based on generative captioning alone typically rely on cross-modal encoders with early fusion of image and text, targeting visual question answering (VQA) and/or captioning tasks with the full encoder-decoder system [63, 22, 10], and do not study the properties of the vision backbone alone.

We revisit image captioning as a pretraining task for learning general vision encoders from web image-text pairs. We compare CLIP-style models with image captioning ones consisting of a Vision Transformer (ViT) [15] encoder and a standard Transformer decoder, henceforth simply referred to as Captioner (Cap). In our experiments, we carefully match pretraining compute and model capacity, and train both models for the same number of examples.

---

[∘]Significant technical contributions. [†]MT led the project.
Code is available at https://github.com/google-research/big_vision.

37th Conference on Neural Information Processing Systems (NeurIPS 2023).

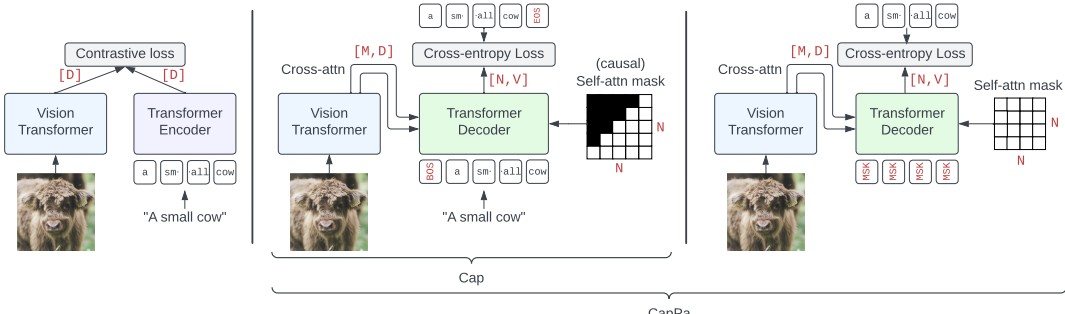

Figure 1: Contrastive models **(left)** use two separate Transformer encoders to extract vector representations from image-text pairs, which are then matched across a potentially large batch [50]. Cap **(middle)** uses a Transformer encoder-decoder architecture [60] and predicts text tokens autoregressively. During training, all tokens are predicted in parallel by shifting the expected output by one token and applying a causal self-attention mask (teacher forcing). In *parallel* decoding **(right)** the Transformer decoder has to predict all tokens at once, conditioned only on the image. CapPa trains a single model switching between autoregressive and parallel decoding.
$D$: model width, $M$: number of image tokens, $N$: number of text tokens, $V$: vocabulary size.

While our results confirm the findings of [50] that Cap models generally lag contrastive ones in zero-shot classification accuracy, the gap becomes smaller with increasing scale. More importantly, the Cap vision backbone matches or outperforms comparable CLIP models in few-shot classification and obtains competitive performance when transferring to classification tasks with large labeled data sets. When combining the vision encoder with a randomly initialized transformer decoder, the Cap pretrained encoder outperforms the CLIP one on captioning, OCR, and VQA tasks. This indicates that a Cap vision backbone might be better suited for multimodal downstream tasks. These benefits can be even more pronounced when combining the vision encoder with a pretrained, partially frozen language decoder similar to [1, 6, 62, 23, 17, 5].

We further propose the CapPa pretraining procedure: the decoder training alternates between standard autoregressive prediction (Cap) and *parallel* prediction (Pa), where the entire caption is predicted in a single decoder forward pass. This only changes the decoder text input (all input tokens are set to [MASK] tokens) and self-attention mask (no mask instead of a causal mask) while requiring no change in the loss or architecture. This mixed training, termed CapPa, significantly improves classification accuracy of the vision backbone.

We ablate how various training and architecture design choices impact performance, and discover promising scaling properties of captioners. Finally, we show that Cap achieves state-of-the-art performance in zero-shot classification tasks which require careful handling of word order and attributes, rather than treating the query as a bag-of-words, as measured by ARO [67] and SugarCrepe [21].

*Overall, our results show that pretraining a simple encoder-decoder architecture via image captioning alone can produce vision encoders competitive with CLIP and presents an interesting alternative to contrastive pretraining—in particular for multimodal vision-language models built form pretrained vision and language modeling components.*

## 2 Related work

Large scale contrastive vision-language pretraining was popularized by CLIP [50] and ALIGN [26]. Several works investigated scaling beyond these works along several relevant axes [49, 69] or using pretrained vision and language backbones with contrastive training [70].

Recent works investigating plain image captioning for pretraining are [13, 54]; [27, 37] study n-gram prediction from images, which can be considered a precursor to captioning. However, image captioning as a pretraining task to learn general vision representations did not attract as much attention as contrastive training, possibly because of inferior and less efficient zero-shot transfer capabilities. Related to captioning, SimVLM [63] uses prefix language modeling to pretrain a multimodal encoder-

decoder model with early vision-language fusion and hybrid convolutional/transformer vision encoder, targeting transfer to VQA and captioning. Further, encoder-decoder models for document, website, and UI understanding are often pretrained to generate captions from rendered websites/documents which trains the model to extract features from those data types and perform OCR [36, 29, 38]. Focusing on image captioning only, LEMON [22] investigates scaling of an encoder-decoder model.

Several works have combined contrastive and captioning losses [66, 40, 34], optionally using a separate text encoder in addition to the decoder. While obtaining excellent results on a range of vision and vision-language tasks, the impact of the loss terms are not well disentangled, nor are compute-matched comparisons of individual losses provided.

Image captioning is a popular ingredient to build large multimodal models from separately pretrained vision and language models [1, 6, 62, 23, 17, 5]. Some of these models freeze large parts of vision and language components, sometimes only learning a small adapter between the two [1, 17, 39]. It is interesting to see how the pretraining strategy for the vision model affects this type of adaptation.

Finally, masked image-text modeling, often combined with image-text matching, is a popular pre-training strategy for encoder-only vision-language models at small data scale [24, 30, 41, 64, 56, 16].

## 3 A simple recipe to pretrain vision encoders via image captioning

Our goal is to establish a pretraining recipe based on image captioning that is comparable to CLIP in terms of simplicity, scalability, and efficiency. Therefore, we adopt Vision Transformer (ViT) [15] backbones as vision encoders, and use a standard Transformer decoder architecture [60] to predict image captions, feeding the ViT-encoded sequence to the decoder via cross-attention. As is common in recent literature [52, 11], we remove biases from attention layers, MLP blocks, and LayerNorms and we replace ReLU by GELU. The decoder input and output embeddings are not shared. Ablations for these choices are in Section 4.4. The decoder has the same width, attention-heads, and MLP dimension as the encoder, but has half the depth. This leads to captioning models which have slightly lower parameter count than corresponding CLIP models, but require about the same pretraining compute in terms of accelerator hours per epoch (see Table 1). We rely on standard next-token-prediction language modeling and train our captioning models with causal attention mask and teacher forcing (Fig. 1, middle), henceforth referring to this variant simply as Captioner (Cap).

**Parallel prediction** We also experiment with a slightly modified training recipe (Fig. 1, right): Instead of training the model only for autoregressively, we train it to predict all tokens in parallel for a fraction of the training examples instead (sampling the prediction type for every example at random throughout training). To this end, we replace the (shifted) decoder input sequence with a sequence of all [MASK] tokens, and drop the causal decoder attention mask. We emphasize that this kind of parallel prediction does not modify the training objective or decoder architecture, and does not incur any extra training cost, but simply modifies the decoder input sequence and attention mask for a fraction of training examples. Moreover, this is different from bag-of-word prediction as not only the presence but also the position of each token has to be predicted. We perform parallel prediction for 75% of the training examples by default and term this variant Cap with parallel prediction (CapPa).

Intuitively, captioning via next token prediction induces an implicit weighting on the supervisory signal of the caption tokens: To predict the first few tokens of a caption, the decoder can benefit a lot from using the image information, while to predict later tokens it can rely more and more on already predicted tokens. Consequently, early tokens might provide a stronger supervisory signal to the encoder than later tokens. By contrast, when predicting all the caption tokens independently in parallel, the decoder can only rely on the image information to predict each token.

**Implementation aspects** We emphasize that our approach closely follows a standard encoder/decoder transformer architecture [60], with the only fundamental difference being the input data format and patch embedding, as well as the modification of the decoder input sequence and attention mask for parallel prediction. This means that our approach is easy to implement in existing transformer code bases [52, 47, 57]. We do not rely on image-specific preprocessing operations other than resizing, see Section 4.1. As a result, existing infrastructure for model-parallel and distributed training can readily be used to scale our approach. In particular, our loss does not require computations across devices the way CLIP does.

Table 2: Performance of frozen visual representations trained via image captioning (Cap/CapPa) and contrastively (CLIP*), when combined with a single transformer decoder trained from scratch for image classification, captioning and VQA (we use CIDEr for captioning, the VQAv2 weighted accuracy for VQAv2, and exact matching accuracy for all other tasks). Bold marks results where two standard-deviation interval overlaps with the two standard-deviation interval of the best result.

| | Classification | | | | | Captioning | | OCR | Question Ans. | |
|---|---|---|---|---|---|---|---|---|---|---|
| | i1k | sun | food | res | pet | COCO | Flickr | VQA | VQAv2 | GQA |
| Cap | $80.2_{\pm0.1}$ | $82.3_{\pm0.2}$ | $90.3_{\pm0.1}$ | $93.6_{\pm0.1}$ | $93.1_{\pm0.1}$ | $\mathbf{117.5_{\pm0.3}}$ | $\mathbf{78.6_{\pm1.1}}$ | $\mathbf{62.2_{\pm0.1}}$ | $\mathbf{68.2_{\pm0.1}}$ | $55.0_{\pm0.1}$ |
| CapPa | $\mathbf{81.3_{\pm0.1}}$ | $82.4_{\pm0.1}$ | $90.9_{\pm0.1}$ | $94.2_{\pm0.2}$ | $\mathbf{94.4_{\pm0.1}}$ | $\mathbf{117.9_{\pm0.6}}$ | $\mathbf{80.5_{\pm0.2}}$ | $\mathbf{62.2_{\pm0.0}}$ | $\mathbf{68.3_{\pm0.1}}$ | $\mathbf{55.7_{\pm0.2}}$ |
| CLIP* (8k) | $81.1_{\pm0.0}$ | $\mathbf{83.2_{\pm0.1}}$ | $91.2_{\pm0.0}$ | $\mathbf{94.8_{\pm0.0}}$ | $93.4_{\pm0.2}$ | $115.8_{\pm0.2}$ | $74.5_{\pm1.2}$ | $56.0_{\pm0.1}$ | $66.5_{\pm0.1}$ | $54.3_{\pm0.3}$ |
| CLIP* (16k) | $\mathbf{81.4_{\pm0.1}}$ | $\mathbf{83.3_{\pm0.1}}$ | $\mathbf{92.0_{\pm0.1}}$ | $\mathbf{95.2_{\pm0.2}}$ | $93.6_{\pm0.2}$ | $\mathbf{116.3_{\pm0.7}}$ | $77.1_{\pm0.7}$ | $56.5_{\pm0.1}$ | $66.7_{\pm0.1}$ | $\mathbf{54.8_{\pm0.6}}$ |
| CLIP | $81.6_{\pm0.1}$ | $82.5_{\pm0.0}$ | $92.6_{\pm0.1}$ | $94.9_{\pm0.1}$ | $93.6_{\pm0.1}$ | $118.4_{\pm0.6}$ | $78.7_{\pm0.9}$ | $60.0_{\pm0.1}$ | $67.8_{\pm0.1}$ | $57.5_{\pm0.1}$ |
| CapPa L/14 | $84.4_{\pm0.0}$ | $84.9_{\pm0.1}$ | $93.8_{\pm0.0}$ | $96.0_{\pm0.1}$ | $\mathbf{95.6_{\pm0.0}}$ | $\mathbf{125.8_{\pm0.1}}$ | $\mathbf{89.3_{\pm1.4}}$ | $\mathbf{65.6_{\pm0.1}}$ | $\mathbf{70.9_{\pm0.0}}$ | $\mathbf{58.3_{\pm0.2}}$ |
| CLIP* L/14 | $\mathbf{84.7_{\pm0.1}}$ | $\mathbf{85.7_{\pm0.1}}$ | $\mathbf{94.6_{\pm0.1}}$ | $\mathbf{96.4_{\pm0.0}}$ | $95.2_{\pm0.1}$ | $123.2_{\pm0.6}$ | $85.5_{\pm0.3}$ | $61.3_{\pm0.2}$ | $68.5_{\pm0.1}$ | $55.3_{\pm0.1}$ |
| CLIP L/14 | $84.8_{\pm0.0}$ | $84.8_{\pm0.1}$ | $95.2_{\pm0.1}$ | $96.3_{\pm0.1}$ | $95.4_{\pm0.3}$ | $124.4_{\pm0.6}$ | $87.1_{\pm0.7}$ | $64.1_{\pm0.0}$ | $70.4_{\pm0.1}$ | $58.7_{\pm0.1}$ |

**Zero-shot classification via scoring**   Image captioning models allow for zero-shot classification simply by scoring the class name. Unlike with contrastive models, where the text (class) embeddings can be computed once and reused for new images, with captioning models all class names have to be scored again for every new image. Cap/CapPa are hence less efficient zero-shot classifiers than CLIP. We emphasize that we focus on learning vision encoders here and zero-shot transfer is not our focus.

## 4   Experiments

### 4.1   Experiment setup

**Pretraining data**   We use a subset of the WebLI data set [6] which contains 10B images and 12B multilingual alt-texts. Specifically, we rely on the WebLI subset corresponding to English websites and apply text-based filtering similar to [26, Sec. 3], [4, Sec 2.2] to obtain 1B image/alt-text pairs, not using any image-text similarity-based filtering. Importantly, WebLI was de-duplicated w.r.t. the images in the evaluation data sets we use in this paper. Please refer to [6, Sec 2.2] for more details on the WebLI data set and to [6, Appendix B] for a datasheet.

Table 1: Parameter count and TPUv4-hrs. per bn. examples seen.

| Model | Params | TPU-hrs. |
|---|---|---|
| B/16 Cap | 192 M | 454 |
| B/16 CLIP* | 197 M | 444 |
| L/14 Cap | 570 M | 1570 |
| L/14 CLIP* | 640 M | 1596 |

**Pretraining details and baselines**[1]   We use a batch size of 8k for our captioning models (larger batch size did not lead to improved performance), and both 8k and 16k for our retrained CLIP baselines (henceforth denoted by CLIP* to avoid confusion with the model checkpoints released by [50], which we reference by CLIP in our results). We explicitly note the batch size for CLIP* models when relevant; CLIP* without modifier refers to the variant trained with batch size 16k. Models are trained on up to 9B image/alt-text pairs (corresponding to 9 epochs on our subset of WebLI). We use the AdaFactor variant from [68] with a cosine schedule (with 10k warmup steps), and set learning rate and decay factor to $10^{-3}$ and $10^{-4}$, respectively. Previous work [70, 69, 59] established these optimizer settings for contrastive pretraining and we adopt them here for captioning. Images are resized to a resolution of $224 \times 224$, and alt-texts are tokenized to a 32k-sized vocabulary with a sentence piece model trained on the English portion of C4 [52], with a maximum sequence length of 64. Following [70, 69, 59] for CLIP* we use identically sized image and text towers, and use global average pooling (GAP) to compute the image representation.

### 4.2   Evaluation protocols and data sets

We focus on properties of the frozen representations and also present some fine-tuning results.

---

[1]Code is available at https://github.com/google-research/big_vision.

Table 3: 10-shot linear evaluation accuracy on the pre-logit representation. CapPa outperforms Cap and achieves overall comparable results with CLIP* trained with a batch size of 16k.

| | INet | CIFAR100 | Pets | Cars |
|---|---|---|---|---|
| Cap | $57.2_{\pm0.1}$ | $58.6_{\pm0.8}$ | $83.7_{\pm0.6}$ | $84.2_{\pm0.1}$ |
| CapPa | $\mathbf{59.1}_{\pm\mathbf{0.1}}$ | $62.4_{\pm0.4}$ | $\mathbf{86.5}_{\pm\mathbf{0.1}}$ | $\mathbf{86.6}_{\pm\mathbf{0.2}}$ |
| CLIP* (8k) | $58.5_{\pm0.1}$ | $64.9_{\pm0.4}$ | $77.7_{\pm1.5}$ | $80.8_{\pm0.4}$ |
| CLIP* (16k) | $\mathbf{59.7}_{\pm\mathbf{0.4}}$ | $66.3_{\pm0.2}$ | $80.6_{\pm1.2}$ | $82.9_{\pm0.1}$ |
| CLIP | $59.0_{\pm0.2}$ | $68.6_{\pm0.1}$ | $82.1_{\pm0.7}$ | $70.2_{\pm0.7}$ |
| CapPa L/14 | $\mathbf{70.6}_{\pm\mathbf{0.2}}$ | $72.9_{\pm0.4}$ | $\mathbf{92.6}_{\pm\mathbf{0.5}}$ | $\mathbf{92.2}_{\pm\mathbf{0.2}}$ |
| CLIP* L/14 | $69.8_{\pm0.1}$ | $\mathbf{74.1}_{\pm\mathbf{0.5}}$ | $87.7_{\pm0.9}$ | $89.2_{\pm0.2}$ |
| CLIP L/14 | $69.7_{\pm0.1}$ | $79.4_{\pm0.3}$ | $90.4_{\pm0.8}$ | $81.1_{\pm0.4}$ |

Table 4: Frozen transfer for zero-shot classification and retrieval via LiT [70]. Especially for the larger model, CapPa is competitve with CLIP* with comparable or fewer LiT examples seen.

| | INet 0shot | | COCO t2i | | COCO i2t | |
|---|---|---|---|---|---|---|
| LiT pairs: | 3B | 12B | 3B | 12B | 3B | 12B |
| Cap | 67.8 | 69.0 | 37.5 | 39.1 | 53.9 | 54.8 |
| CapPa | 68.8 | **70.2** | 37.3 | 38.6 | 53.9 | 55.1 |
| CLIP* | 69.0 | 70.0 | 38.9 | **40.1** | 55.1 | **57.0** |
| CLIP | 68.3 | | 32.3 | | 52.8 | |
| CapPa L/14 | 76.4 | **77.5** | 43.9 | 45.4 | 60.3 | **62.6** |
| CLIP* L/14 | 75.8 | 76.6 | 44.7 | **46.3** | 60.7 | 62.3 |
| CLIP L/14 | 75.1 | | 36.5 | | 56.6 | |

**Probing the frozen representation**  As an inexpensive way to assess classification performance we use the linear adaptation protocol (based on the pre-logits layer for CLIP* and GAP of the encoder output sequence for our captioning models) and eval sets from [68, 70], reporting the 10-shot classification accuracy. We also evaluate the classification accuracy when using the full ImageNet1k training set to learn a dense layer, an MLP, and a multihead attention pooling (MAP)-based [35] classifier. To assess the amenability of the different the CLIP* and CapPa vision encoders to fine-grained classification, we train MAP-based classifiers on a range of specialized data sets which can be divided in two groups. The first group requires fine-grained classification of animal or plant breed [45, 61, 48, 28], or product variant [32, 3], whereas data sets in the second group covers a range of distinct objects [18, 12, 8, 71, 46, 19].

**Text encoder/decoder-based inference**  We use LiT [70] to learn a text embedding matched to the embedding of our pretrained vision encoders. Generally, LiT is an efficient way to equip any pretrained vision backbone with zero-shot classification and retrieval capabilities, here particularly for Cap whose pretrained decoder does in principle have these capabilities but incurs significant inference cost. We follow the setup from [69] and assess the zero-shot classification accuracy on ImageNet1k [53] and retrieval recall@1 on COCO captions [7]. We choose the LiT text encoder to mirror the architecture of the vision encoder at hand and attach a randomly initialized MAP head to the vision encoder to map into the shared image-text embedding space. For comparison, we also apply LiT to our CLIP* models; this is not necessarily meant to be practically useful (continuing training the CLIP* model might be a better investment of compute).

Motivated by recent work combining pretrained vision backbones and language models [1, 6, 62, 23, 17, 5], we investigate the amenability of the learned representations to interface with a text decoder. Concretely, we use the "LiT decoder" setup [2] which trains a transformer decoder from scratch on top of a frozen vision representation to solve captioning [7, 65], VQA [20, 25, 43] and classification [53, 71, 3, 8, 48] tasks in a multitask fashion (we use the default hyperparameters from [2] except for the data mixing strategy set to "concat image-question pairs" [2, Sec. 5.3] ).

In addition, we explore combining our representations with a pretrained T5 decoder [52]. We rely on the previously described multitask LiT decoder setup and tune the most important hyper parameters (see supplementary material for details). For the T5 decoder we keep all the parameters frozen but reinitialize and train the cross-attention layers. Finally, we also leverage a frozen, pretrained 12-layer GPT-2 decoder [51] for image captioning by combining it with the frozen vision encoders via an adapter, similar to ClipCap [44].

**Fine-tuning**  We fine-tune our vision encoders on the full ImageNet1k data set, attaching a randomly initialized MAP head to the pretrained representation (see supplementary material for details).

**Using the pretrained text decoder**  Finally, we also evaluate our models (with the pretrained text decoder) on the SugarCrepe [21] and the Attribute, Relation and Order (ARO) [67] benchmarks (which are derived from different captioning data sets [33, 65, 7]). Specifically, SugarCrepe and ARO shuffle the attributes, relations and order of captions and measures the sensitivity of vision-language

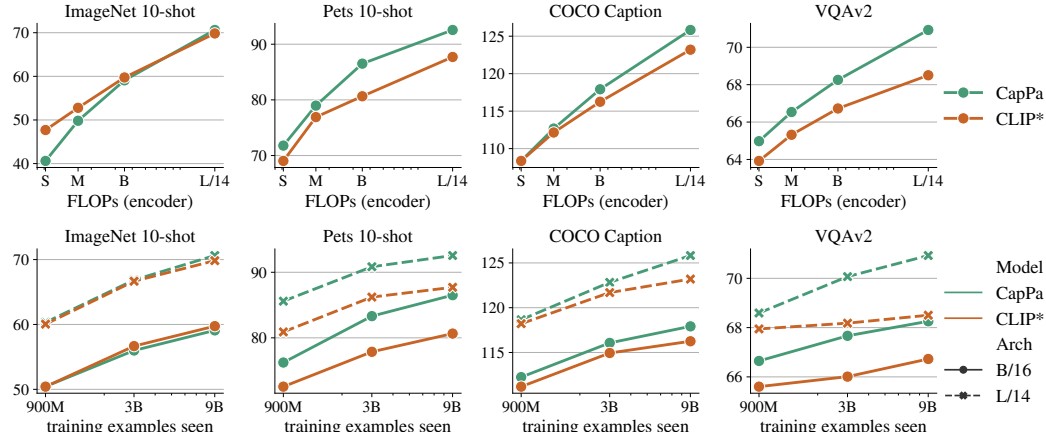

Figure 2: 10-shot classification accuracy on the frozen pre-logit representation (left two columns); captioning and VQA performance with a decoder (right two columns). **Top row:** Performance of vision backbones pretrained with captioning (Cap/CapPa) and contrastively (CLIP*) as a function of the model size/FLOPs (we compare ViT-S/16, M/16, B/16, and L/14). CapPa exhibits favorable scaling behavior on captioning and VQA tasks. **Bottom row:** Performance of CapPa and CLIP* as a function of the number of training examples seen. The behavior is similar as for model scale.

models to these manipulations. As shown by [67], contrastive models are not very sensitive to precise relational and attributional information and behave closer to bag-of-words models. Intuitively, since captioning-based models model the joint distribution over all tokens, it is interesting to see how they compare to contrastive models on ARO. For each example, we use log-likelihood to score both the true caption and the shuffled captions. The model prediction is the caption with the highest score.

## 4.3 Main results

Tables 2, 3, 4, and Fig. 3 show the LiT decoder results, the 10-shot classification accuracy, the LiT tuning results, and the full linear probing accuracy for our models trained for 9B examples seen.

CapPa outperforms Cap across almost all evaluations, and often also outperforms CLIP* trained with the same batch size (8k), while being competitive with CLIP* trained with a batch size of 16k. This is remarkable since CLIP* benefits substantially from a large batch size [69]. The trend becomes more pronounced when increasing the model size: CapPa clearly outperforms CLIP* in 10-shot classification accuracy for a ViT-L/14 encoder (Table 3).

Table 5: Finetuning on INet.

|  | B/16 | B/16$_{384}$ | L/14$_{336}$ |
| --- | --- | --- | --- |
| CLIP* | 84.9 | 86.0 | 88.1 |
| CapPa | 84.4 | 85.7 | 87.7 |
| Cap | 83.9 | 85.3 | - |

When transferred to ImageNet1k classification with a linear probe (Fig. 3), the frozen Cap and CapPa encoders lag somewhat behind CLIP*, but the gap almost vanishes when using a MAP head instead of a linear probe. This is not very surprising, given CLIP* models are trained with a linear head, which might induce linear separability in the average pooled encoder output representation. In contrast, the Cap models feed into a decoder via cross-attention which might not impose linear separability as strongly. For fine-tuning the full model on ImageNet1k (Table 5) we only observe a minor gap between CapPa and CLIP*. As for text encoders learned

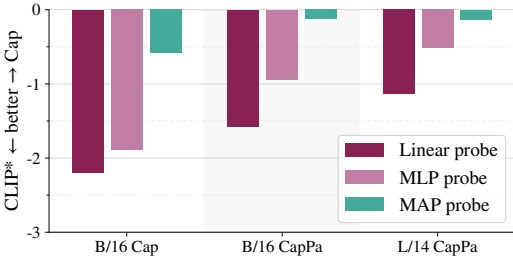

Figure 3: Linear probing makes cap pre-trained image encoders seem worse, but when learning the pooling (MAP probe), the gap is essentially closed.

via LiT (Table 4) CapPa outperforms CLIP* for long LiT training and large model size in zero-shot classification, but is outperformed by CLIP* in retrieval. Notably, for a short LiT tuning with 3B examples our models outperform CLIP [50] in zero-shot classification and retrieval while matching

| | Attrib. | Rel. | Order (F) | Order (C) |
|---|---|---|---|---|
| Blind dec. | 83.7 | 86.2 | 98.8 | 98.7 |
| Cap | **88.9** | 86.6 | 99.1 | **99.0** |
| CapPa | 85.7 | **86.7** | **99.2** | 98.8 |
| CLIP* | 53.2 | 39.7 | 45.5 | 37.0 |
| CLIP | 62.7 | 58.7 | 57.9 | 49.5 |
| ARO Best | 88.0 | 73.0 | 60.0 | 46.0 |
| NegCLIP | 71.0 | 81.0 | 91.0 | 86.0 |

Table 6: Results on the Attribute, Relation and Order (ARO) benchmark [67]. Cap and CapPa clearly outperform all CLIP* and CLIP variants across all data sets. They also outperform Neg-CLIP [67] which was explicitly trained to be sensitive to word ordering and attribution.

| Training | Arch | Repl. | Swap | Add |
|---|---|---|---|---|
| Cap | B/16 | **88.21** | **84.00** | 98.94 |
| CapPa | B/16 | 87.67 | 83.11 | **99.13** |
| CLIP* | B/16 | 81.95 | 63.22 | 81.91 |
| NegCLIP | B/32 | 85.36 | 75.33 | 87.29 |
| OpenCLIP | G/14 | 86.50 | 68.56 | 88.36 |

Table 7: Results on the SugarCrepe [21] benchmark, which fixes known issues in previous image-text benchmarks like ARO. Full results are in Table 18 in the appendix. Small Cap and CapPa models outperform even large or hard-negative trained CLIP models in all categories, and essentially solve the category of tests which "Add"s plausible but wrong details to the caption.

the number of examples seen by CLIP (12.8B) when summing over pretraining (9B) and LiT tuning (3B), despite the vision encoder being frozen during LiT tuning (which saves compute). Matching the number of examples seen by CLIP during LiT tuning (12B) leads to a clear additional improvement.

Combining our models with a fresh (LiT) decoder (Table 2), we observe that CapPa performs better than CLIP* trained with batch size 16k on captioning and VQA, while obtaining competitive accuracy on classification tasks. Again, this pattern becomes more pronounced with increased models size. Indeed, for a ViT-L/14 encoder CapPa even outperforms CLIP in the majority of LiT decoder tasks.

**Scaling properties** In Fig. 2 we present an analysis of our encoders as a function of the model size and the number of training examples seen for a selection of 10-shot classification and vision & language evaluations using a fresh decoder (plots for all evaluations and Cap can be found in the supplementary material). It can be seen that CLIP* and CapPa models exhibit similar scaling behavior, with CapPa showing a somewhat steeper increase in captioning and VQA performance as a function of model size and examples seen, in particular for a ViT-L/14 backbone. This indicates that the benefits of CapPa models might become more pronounced with further model and data scaling.

**Attribution, relation, ordering** Table 6 presents our results on the ARO Benchmark [67]. Cap and CapPa models achieve close to a perfect score on the ordering subsets. They outperform the best contrastive variants by around 30% and 40% on Flickr Order and COCO Order, respectively. We also compare with NegCLIP [67], which employs additional supervision to make contrastive models sensitive to word ordering and attribution. Cap and CapPa exceed NegCLIP by 8% and 13% out-of-the-box on COCO Order and Flickr Order. The same can be observed for the attribution, and relation subsets of ARO. So we are facing a clear trade-off: CLIP-style models outperform captioning-based models in terms of standard zero-shot classification accuracy, whereas the latter are much better at processing fine-grained descriptions of visual content. Finally, we also trained

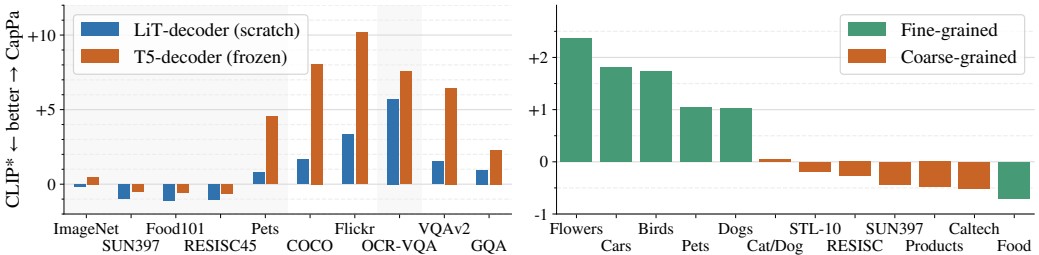

Figure 4: Absolute improvement of CapPa over CLIP* in various settings. **Left:** CapPa pairs significantly better with decoders in image-language tasks, especially when the decoder is a pre-trained and frozen language model. **Right:** CapPa seems to be a noticeably better frozen feature extractor for fine-grained classification tasks (we show L/14 results, see appendix for B/16).

a Cap model with no image input (Blind dec.), which is just a language model for alt-text. This model overall performs only slightly worse than Cap and CapPa, which suggests that the sentence manipulation in ARO can to a large extent be detected by language modeling alone. This was also observed in concurrent work [42, 21].

**SugarCrepe**   The SugarCrepe benchmark [21], introduced concurrently to our work, promises to fix the issues in ARO and similar benchmarks; for instance, a blind model is no better than chance. Still, even our small captioning pretrained B/16 model outperforms the largest G/14 contrastive model as well as the specifically trained NegCLIP. The "Swap" category is especially sensitive to the relation between multiple things in the image, something that is fundamentally hard for contrastive pretraining to learn. The "Add" category, where highly plausible but wrong things are added to the text is essentially solved by captioners. The full breakdown, more baselines, and qualitative examples are provided in Tables 18 and 22–24 in the appendix. This result is strong evidence that captioning as a pretraining objective imbues capabilities to the model that contrastively trained models are blind to.

**Frozen T5 decoder**   We also trained models with frozen encoder *and* frozen decoder (initialized with a T5 checkpoint). For these experiments, only the cross attention weights were trained (28M trainable parameters, compared to the 248M trainable parameters when the decoder is trained from scratch). The relative improvements of CapPa over CLIP* are shown in Fig. 4 (left). Even though the absolute performance of the decoder trained from scratch (Table 2) is better than when only training the cross attention weights (Table 12), CapPa with a frozen decoder closes the gap on three classification tasks, reverts the trend on ImageNet, and improves the performance by large margins on Pets, as well as all captioning and VQA tasks. This result suggests that the captioning objective is better suited to train an encoder that is later combined with a pretrained language decoder.

**Frozen GPT-2 decoder**   We combined our frozen vision encoders with a frozen GPT-2  model from [51] using an adapter as proposed by ClipCap [44]. We found that this setup is less well suited for VQA and multitask conditioning, so we only evaluate it on captioning in single-task fashion as done by ClipCap. We tried the MLP and transformer adapters from [44], but obtained better results for both CLIP* and CapPa with a "resampler", a LayerNorm followed by a single cross-attention layer with learnable query embeddings, generating a prefix of 32 soft tokens for GPT-2. This resampler has only 7.1M parameters, about $4\times$ less than the MLP adapter from [44]. CapPa still outperforms CLIP* in this setup, but the gains are more modest than for the T5 decoder and a decoder trained from scratch (both single-task, see Fig. 5). We emphasize that both encoder and GPT-2 decoder are frozen and we only use a small adapter.

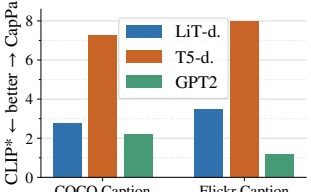

Figure 5: Absolute improvement (single-task) of CapPa over CLIP* for a decoder trained from scratch (LiT-d.), a frozen T5 decoder, and a frozen GPT-2 similar to [44].

**Fine-grained classification**   Fig. 4 (right) shows the improvement when using CapPa instead of CLIP* for a range of specialized data sets. CapPa outperforms CLIP* on the majority of fine-grained tasks which suggests that captioning as a pretraining task leads to better features for fine-grained classification than contrastive pretraining.

## 4.4   Ablations

All models discussed in this section are trained for 900M training examples seen.

**Parallel prediction**   Recall that for parallel prediction we replace all text input tokens with `[MASK]` tokens. An alternative would be to only replace a random subset, as done e.g. in [22], to provide a partial context for the prediction. However, we did not observe improvements of the vision encoder when only masking a fraction of the tokens, so we focused on fully masked input sequences. For fully masked input sequence Fig. 6 (left) shows the improvement in 10-shot classification accuracy over training for pure autoregressive decoding as a function of the fraction of examples for which the decoder is trained for parallel prediction instead. A fraction of 0.75 leads to a balanced improvement across all considered data sets. Finally, alternating between parallel and autoregressive prediction for all examples in a batch, rather than performing parallel prediction with mixed batches, led to significantly worse results.

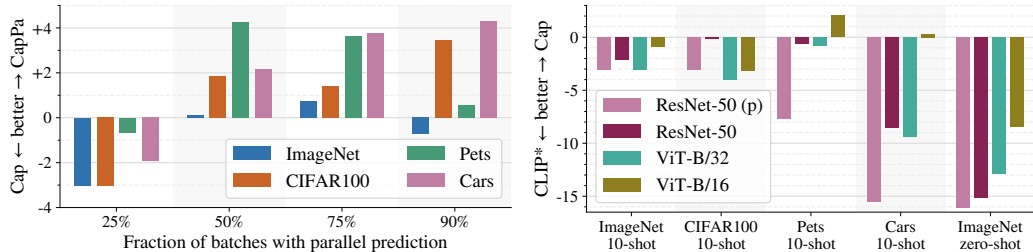

Figure 6: **Left:** Improvement in 10-shot classification accuracy as a function of the fraction of training examples for which parallel prediction is performed in CapPa. A fraction of 0.75 leads to a balanced improvement. **Right:** Improvement of 10-shot/zero-shot (without prompts) classification accuracy when using Cap instead of CLIP*. For ResNet-50 (p) the decoder consumes 4 averaged image tokens as a prefix (no cross-attention). Cap is competitive in 10-shot accuracy for ViT-B/16.

**Encoder architecture**  Next, we investigate the effect of the encoder architecture on the representation quality, comparing CLIP* with Cap when using a BiT ResNet50 [31], ViT-B/32, and a ViT-B/16 vision encoder. We use a B-sized text encoder and B-sized 6-layer decoder for CLIP* and Cap, respectively, for all encoders, except for ResNet50 for which we also train with a prefix decoder following [50, Sec. A2]. According to [15] the BiT ResNet50 obtains performance roughly comparable to ViT-B/32 when trained and evaluated on image classification. Further, ResNet50 and ViT-B/32 both produce a sequence of length 49 before pooling for $224 \times 224$ images, which we feed to the decoder. Fig. 6 (right) shows the improvement obtained when using Cap instead of CLIP* as a function of the encoder architecture. The improvement (regression) in ImageNet zero-shot accuracy is smallest (largest) for the ResNet50 architecture (this is the architecture used to create [50, Fig. 2] that compares contrastive with bag-of-words and captioning approaches) and is significantly improved when using a ViT architecture and when reducing the patch size (which does not increase model capacity but the sequence length and hence FLOPs). Also recall that these models were only trained on 900M examples. Interestingly, the difference between CLIP* and Cap on 10-shot metrics are generally smaller, and for a ViT-B/16 encoder the two approaches lead to similar performance.

**Captioning task variants**  We train Cap while randomly reversing the caption with probability 0.5. This maintains model capacity and pretraining compute unchanged. We do not observe improved performance (see Table 8, left). While [13] ablates backwards captioning and shows improvements, they use a separate decoder for backwards captioning, so the ablated model has fewer parameters and FLOPs (here we control for both factors). Additionally, we train a CapPa variant with two parallel decoders, one for autoregressive prediction and another one for parallel prediction, each with 3 layers (instead of 6). This model matches the pretraining compute of the default CapPa but underperforms in the majority of 10-shot tasks.

**Training with language-pretrained decoder**  Finally, we train Cap ViT-B/16 with a frozen pretrained T5-Base decoder (which has 12 decoder layers). To obtain a stable training setup we adapt the optimizer hyper-parameters (learning rate 0.0005, $\beta_2 = 0.95$) and unfreeze the cross attention. Optionally, we re-initialize the cross-attention parameters and unfreeze the decoder. None of these variants performs better than Cap overall (see Table 8, right), and the more we allow the decoder to deviate from its language-pretrained weights the better the vision performance gets.

Table 8: **Left:** Comparing Cap and CapPa with other captioner variants: Forward and backward captioning with the same decoder, autoregressive and parallel prediction with two 3-layer decoders. **Right:** Training Cap with a frozen T5 decoder does not help, even when unfreezing parts of it.

| | ImageNet | CIFAR100 | Pets | Cars | | ImageNet | CIFAR100 | Pets | Cars |
|---|---|---|---|---|---|---|---|---|---|
| Cap | 49.7 | 56.0 | 72.6 | 74.7 | Cap (12 layers) | 48.7 | 54.8 | 74.4 | 73.8 |
| Cap (fw.+bw.) | 49.2 | 56.1 | 71.7 | 73.0 | + frozen T5 dec. | 42.8 | 44.9 | 69.3 | 62.3 |
| CapPa 2 dec. | 49.5 | 54.9 | 75.8 | 79.0 | + reinit. xatt. | 43.7 | 45.7 | 68.3 | 66.9 |
| CapPa | 50.4 | 57.4 | 76.2 | 78.5 | + unfreeze dec. | 48.6 | 55.2 | 72.0 | 75.6 |

**Decoder architecture** While following the original transformer decoder architecture [60] closely, we adopt the now common change of removing the biases in the decoder [52, 11] to improve stability without affecting performance, see Table 19 (left) in the appendix. We use separate embeddings for the decoder's input and output, Table 19 (left) shows that this works slightly better. We use six decoder layers (see Table 19, right) which simultaneously leads to overall good results while also matching the total parameter count of the corresponding CLIP* model.

Inspired by experiments from [68], where applying a stronger weight decay to the classification head of ViTs trained for image classification led to accuracy improvements, we also experimented with increased weight decay applied to the decoder or cross-attention layers, but we did not observe any benefits from this. Further, we explored using a tokenizer trained on 300M randomly sampled WebLI alt-texts instead of the one pretrained on C4, which did not improve accuracy.

**Effect of pretraining data** So far all our results were based on models pretrained on a variant of WebLI, and one might wonder whether our findings transfer to pretraining on other data sets. We therefore train some of our models and baselines on the smaller, publicly available LAION-400M dataset [55] which was collected and filtered following a different protocol. For instance, it was filtered using an existing CLIP model to score image-text pairs, which might induce a significantly different bias in the training data and our conclusions. However, the 10-shot linear classification results in Fig. 7 show that the conclusions remain the same: CapPa achieves accuracy comparable with CLIP* and outperforms Cap.

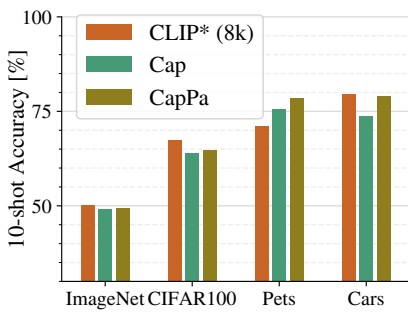

Figure 7: Pretraining on LAION-400M.

## 5   Discussion and conclusion

We presented an extensive comparison of vision encoders pretrained with a contrastive and generative (captioning) objective and found that the generatively pretrained encoders obtain better performance when used for captioning, VQA, fine-grained and few-shot classification tasks, while achieving competitive performance in classification overall. This is in contrast to previous work [50] which argued that predicting alt-text from images *explicitly and exactly* is an overly challenging pretraining task and may lead to sub-par vision capabilities. Moreover, our results show that captioning as a pretraining task might exhibit favorable scaling properties with increasing model and data scale, and we hope future work will explore this avenue.

One downside of our approach is that the Cap/CapPa text decoder is of limited use. While it achieves excellent results when word order and object relationships matter—a scenario where CLIP-style models exhibit a strong bag-of-word bias—zero-shot classification and retrieval are computationally expensive. We showed that this can be mitigated relatively cheaply by training a text encoder matched to the frozen Cap/CapPa representation via LiT tuning. Note that such a procedure is cheaper than training the text encoder with the image encoder and text decoder throughout the whole pretraining procedure as done e.g. by CoCa [66].[2] Another promising avenue is to explore parallel prediction for inference task, as it would allow for efficient scoring e.g. for zero-shot classification. Indeed, parallel prediction produces a joint distribution over all tokens, which can be used to score an arbitrary number of classes or queries.

In conclusion, we established plain image captioning as a competitive pretraining strategy for vision backbones from image-text data. We hope to inspire follow up work dedicating more attention the advantages of captioning as a pretraining task for vision encoders.

---

[2][66, Table 8b)] presents an ablation of the contrastive loss in CoCa, but keeps the unimodal decoder part (without cross-attention) which serves as text encoder for the contrastive loss, and hence obtains essentially no gains in accelerator time. The Cap/CapPa decoder, in contrast, does not have a unimodal component, and is hence cheaper to evaluate than the CoCa decoder for given architecture size (base, large).

**Acknowledgments** We would like to thank Jannis Bulian, Mostafa Dehghani, Alexey Dosovitskiy, Daniel Keysers, Mario Lucic, Basil Mustafa, and Xiao Wang for feedback on this paper and inspiring discussions on captioning-based pretraining. We would also like to thank Janko Ferlič from Unsplash for providing the photo used in Fig. 1.

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

# A  Transfer and finetuning details

**Few-shot evaluation**   We use the linear adaptation protocol and evaluation sets from [68, 70], reporting the 10-shot classification accuracy. Specifically, we rely on the pre-logits layer for CLIP* and GAP of the encoder output sequence for our captioning models. For every combination of data set and model we run the 10-shot adaptation three times and report the mean (and standard deviation for key results).

**LiT decoder and T5 decoder**   To train a multi-task decoder from scratch on top of the frozen representation for classification, captioning and VQA, we precisely follow the setup and hyper parameters from [2] except for the data mixing strategy, for which we set to "concat image-question pairs" ([2, Sec. 5.3]). For all encoders, we use the full feature sequence before pooling (including the class token for the evaluation of CLIP). Throughout, we rely on a B-sized transformer decoder [60] with 12 layers.

We also tried fine-tuning the image encoder along with the decoder for both CLIP* and Cap/CapPa models and did not obtain an improvement for any of the models. This is consistent with prior work which did not observe an improvement either for CLIP-style models when fine-tuning with the same decoder-based setup, see [2, Sec. 5.7].

For the T5 decoder we keep all the parameters frozen but reinitialize and train the cross-attention layers. We perform a small sweep around the default learning rate and weight decay of the setup used for training from scratch, while keeping the other hyperparameters unchanged.

**Linear and non-linear ImageNet-1k probes (frozen transfer)**   When performing linear and non-linear probes on ImageNet-1k, we run a wide hyper-parameter optimization sweep for all types of probes (linear, MLP, MAP) in order to get solid, trustworthy conclusions. Specifically, for each image encoder and probe combination, we sweep the full cross-product over the following hyper-parameters: **epochs**: (1, 3, 10, 30, 100); **image cropping**: `resize(256)|random_crop(224)` or `inception_crop(224)`; **learning rate**: (0.001, 0.0003, 0.0001) plus earlier runs showing 0.003 and 0.01 to perform much worse; **weight decay**: (0.0001, lr * 0.1, 0.0); **hidden dimension**: 0 or 1024; **loss**: sigmoid or softmax cross-entropy. The head weights are always initialized to 0 and its bias to -6.9 in the sigmoid case.

For each result shown in Fig. 3, we select the best setting using 1% of the training data that was held-out for this purpose, and report its accuracy on the 50 000 images in the validation set. For completeness, we further compute various ImageNet test-set variants and report full results in Table 9.

**Broad MAP-head transfers (fine-grained)**   We run the same sweep as described above for each individual dataset and model combination, but only using the MAP-head probe. For each dataset, we either use a provided held-out validation set for selecting the best settings, or hold out 20% of the training set if none is provided. Full numeric results are provided in Table 10. Note that we selected and classified the datasets as coarse- or fine-grained solely by looking at the datasets and their classes, before running any single experiment on them, and never revisited this selection.

**Fine-tuning on the full ImageNet-1k data set**   When fine-tuning on the full ImageNet-1k dataset, we attach a fresh MAP head to the pretrained encoder and run full fine-tuning using the AdaFactor optimizer modified for ViTs in [68]. In each setting (B/16, B/16$_{384}$, L/14$_{336}$), we run the exact same sweep for CLIP*, CapPa, and Cap models. Notably, our exploration is significantly smaller than that of [14] and unlike for CLIP [50], ImageNet was fully de-duplicated from our pre-training dataset. In all cases, we select the best model on a held-out 2% of the training data and report that model's performance on the 50 000 image validation set without re-training.

For the B/16 models, we sweep over three **learning rates**: (0.0001, 0.00003, 0.00001); two **layer-wise learning-rate decays**: (None, 0.8); 2 **RandAugment** parameters: (10, 15); 3 **Mixup**: (0.0, 0.2, 0.5); and five **Polyak (EMA) averaging factors**: (None, 0.9, 0.999, 0.99999, 0.9999999) with a batch size of 2048 and 100 epochs. The best setting uses learning rate 0.00001, layer-wise decay 0.8, Mixup 0.5 and no Polyak averaging.

For the L/14 models at 336 px resolution, we sweep over three **learning rates**: (0.001, 0.0003, 0.0001), three **layer-wise learning-rate decays**: (None, 0.9, 0.8), and five **Polyak (EMA) averaging factors**:

(None, 0.9, 0.999, 0.99999, 0.9999999). Note that the latter does not require re-training for each setting and hence is cheap. We fix rand-augment to (2, 10), Mixup to 0.2, and training duration to 50 000 steps with batch-size 512, without revisiting these choices. Besides that, we mostly follow [15, 68]. The best setting uses learning rate 0.0001, layer-wise decay 0.9, and Polyak 0.99999 for both models.

## B Additional Results

### B.1 Probing and LiT tuning results

Table 9 shows the classification accuracy on different ImageNet-1k evaluation sets, when probing the frozen representation with different probes (linear and non-linear), extending the numerical results from Fig. 3.

Table 10 presents transfer results of the frozen representation to fine- and coarse-grained classification tasks (using a MAP head). This complements the results from Fig. 4 (Right).

Table 11 expands Table 4 in the main paper and shows frozen transfer for zero-shot classification and retrieval via LiT [70].

Table 12 presents the performance of frozen Cap/Cap and CLIP* encoders when combined via cross-attention with a *frozen* T5 decoder. This represents the data from Fig. 4 (Left) in the main paper in tabular form.

Table 9: Extended numerical results for Fig. 3, i.e. linear and non-linear ImageNet-1k probes on top of the frozen models. While the *linear* separability of CLIP models is higher, the gap between CLIP* and Cap models is mostly closed when the probe also learns how to pool (*map*).

| Model | Head | Top-1 | ReaL | -v2 | -R(endition) | -A(dvers.) | ObjectNet |
|---|---|---|---|---|---|---|---|
| CLIP* (8k) | linear | 79.8 | 85.6 | 69.0 | 71.9 | 38.0 | 49.8 |
| | mlp | 80.4 | 86.1 | 69.6 | 74.4 | 39.3 | 50.9 |
| | map | 82.2 | 87.3 | 71.5 | 72.9 | 34.3 | 49.0 |
| | map+mlp | 82.2 | 87.4 | 71.7 | 71.8 | 34.8 | 48.3 |
| CLIP* (16k) | linear | 80.2 | 85.9 | 69.2 | 73.2 | 40.3 | 51.3 |
| | mlp | 80.9 | 86.1 | 70.3 | 71.4 | 37.3 | 49.8 |
| | map | 82.6 | 87.5 | 72.4 | 73.9 | 37.2 | 50.0 |
| | map+mlp | 82.6 | 87.5 | 72.1 | 73.0 | 36.3 | 49.3 |
| Cap | linear | 77.7 | 84.1 | 67.1 | 68.2 | 24.1 | 44.2 |
| | mlp | 78.5 | 84.8 | 68.0 | 76.0 | 27.1 | 45.6 |
| | map | 81.6 | 87.0 | 71.3 | 76.2 | 32.3 | 45.8 |
| | map+mlp | 81.5 | 87.0 | 71.5 | 76.2 | 31.4 | 45.8 |
| CapPa | linear | 78.3 | 84.6 | 66.5 | 67.7 | 22.1 | 43.7 |
| | mlp | 79.4 | 85.6 | 68.6 | 77.2 | 25.6 | 46.2 |
| | map | 82.0 | 87.5 | 72.3 | 80.9 | 41.5 | 50.1 |
| | map+mlp | 82.1 | 87.3 | 72.0 | 79.4 | 39.1 | 49.5 |
| CLIP* L/14 | linear | 84.2 | 88.4 | 75.0 | 83.8 | 59.1 | 60.2 |
| | mlp | 84.6 | 88.5 | 74.9 | 83.3 | 56.6 | 58.6 |
| | map | 85.9 | 89.3 | 76.7 | 84.9 | 57.4 | 58.2 |
| | map+mlp | 85.8 | 89.2 | 77.0 | 83.6 | 56.1 | 57.8 |
| CapPa L/14 | linear | 83.0 | 87.7 | 73.1 | 81.1 | 41.6 | 53.8 |
| | mlp | 84.1 | 88.7 | 74.6 | 87.3 | 47.0 | 56.8 |
| | map | 85.8 | 89.3 | 76.8 | 86.1 | 54.5 | 56.6 |
| | map+mlp | 85.8 | 89.2 | 76.6 | 85.5 | 52.2 | 56.5 |

Table 10: Transfer of the frozen representation to fine- and coarse-grained classification tasks (using a MAP head). This extends the numerical results of Figure 4 (Right).

| Dataset | Grain | CLIP* (8k) | CLIP* (16k) | Cap | CapPa | CLIP* L/14 | CapPa L/14 |
|---|---|---|---|---|---|---|---|
| Dogs [28] | Fine | 77.5 | 77.9 | 79.6 | 81.2 | 85.0 | 86.0 |
| Flowers [45] | Fine | 85.1 | 89.5 | 94.0 | 97.0 | 96.6 | 98.9 |
| Birds [61] | Fine | 76.9 | 78.1 | 76.3 | 54.2 | 85.0 | 86.7 |
| Pets [48] | Fine | 91.2 | 91.2 | 91.5 | 94.4 | 94.4 | 95.4 |
| Cars [32] | Fine | 90.8 | 91.6 | 93.4 | 93.3 | 94.0 | 95.8 |
| Food [3] | Fine | 91.0 | 92.1 | 91.1 | 91.5 | 94.9 | 94.2 |
| RESISC [8] | Coarse | 92.5 | 96.4 | 90.5 | 96.1 | 97.1 | 96.9 |
| Products [46] | Coarse | 88.8 | 89.0 | 87.8 | 88.5 | 90.7 | 90.3 |
| SUN397 [71] | Coarse | 81.8 | 82.9 | 82.0 | 82.1 | 85.7 | 85.2 |
| Caltech [19] | Coarse | 93.5 | 93.1 | 89.0 | 86.5 | 93.8 | 93.2 |
| STL-10 [12] | Coarse | 98.0 | 98.5 | 97.7 | 98.1 | 99.2 | 99.1 |
| Cat/Dog [18] | Coarse | 99.9 | 99.9 | 99.7 | 99.7 | 99.8 | 99.9 |

Table 11: Frozen transfer for zero-shot classification and retrieval via LiT [70] as a function of the number of number of training examples seen by the text encoder (the vision encoder is pretrained and frozen, and equipped with a MAP head which is trained along with the text encoder). The text encoder mirrors the architecture of the vision encoder. Especially for the larger model, CapPa is competitve with CLIP* with comparable or fewer examples seen. The CLIP numbers are obtained by evaluating the image and text encoders released by [50] in our eval setup. We report these numbers for reference, no LiT tuning is done on top of the CLIP vision encoder. This table complements Table 4 in the main paper.

| | ImageNet 0shot | | | | COCO t2i r@1 | | | | COCO i2t r@1 | | | |
|---|---|---|---|---|---|---|---|---|---|---|---|---|
| LiT pairs: | 0 | 900M | 3B | 12B | 0 | 900M | 3B | 12B | 0 | 900M | 3B | 12B |
| Cap | - | 65.9 | 67.8 | 69.0 | - | 35.3 | 37.5 | 39.1 | - | 50.3 | 53.9 | 54.8 |
| CapPa | - | 66.4 | 68.8 | 70.2 | - | 34.3 | 37.3 | 38.6 | - | 49.7 | 53.9 | 55.1 |
| CLIP* (8k) | 65.6 | 65.9 | 67.6 | 69.0 | 41.5 | 36.5 | 38.2 | 39.5 | 56.7 | 52.0 | 54.0 | 56.1 |
| CLIP* (16k) | 67.7 | 66.7 | 69.0 | 70.0 | 43.0 | 37.0 | 38.9 | 40.1 | 58.2 | 53.0 | 55.1 | 57.0 |
| CLIP | 68.3 | | | | 32.3 | | | | 52.8 | | | |
| CapPa L/14 | - | 74.6 | 76.4 | 77.5 | - | 40.6 | 43.9 | 45.4 | - | 56.6 | 60.3 | 62.6 |
| CLIP* L/14 | 74.8 | 74.5 | 75.8 | 76.6 | 48.1 | 42.7 | 44.7 | 46.3 | 63.7 | 57.7 | 60.7 | 62.3 |
| CLIP L/14 | 75.1 | | | | 36.5 | | | | 56.6 | | | |

Table 12: Performance of frozen representations trained via image captioning (Cap/CapPa) and contrastive (CLIP*) objective, when combined via cross-attention with a *frozen* T5 decoder. Only the cross-attention weights are updated during the training. See Table 2 for the corresponding models that have the decoder trained from scratch.

| | Classification | | | | | Captioning | | OCR | Question Ans. | |
|---|---|---|---|---|---|---|---|---|---|---|
| | i1k | sun | food | res | pet | COCO | Flickr | VQA | VQAv2 | GQA |
| Cap | $79.0_{\pm0.1}$ | $81.3_{\pm0.1}$ | $89.3_{\pm0.0}$ | $92.4_{\pm0.1}$ | $92.3_{\pm0.3}$ | $119.7_{\pm0.6}$ | $72.2_{\pm0.9}$ | $57.7_{\pm0.0}$ | $64.6_{\pm0.1}$ | $52.1_{\pm0.2}$ |
| CapPa | $80.0_{\pm0.0}$ | $81.2_{\pm0.1}$ | $89.9_{\pm0.0}$ | $93.1_{\pm0.2}$ | $93.2_{\pm0.3}$ | $118.7_{\pm0.5}$ | $70.0_{\pm0.5}$ | $57.8_{\pm0.2}$ | $63.3_{\pm0.3}$ | $51.9_{\pm0.3}$ |
| CLIP* (8k) | $79.1_{\pm0.0}$ | $81.5_{\pm0.2}$ | $89.9_{\pm0.0}$ | $92.7_{\pm0.2}$ | $88.5_{\pm0.2}$ | $110.6_{\pm0.5}$ | $60.8_{\pm1.0}$ | $50.3_{\pm0.3}$ | $57.2_{\pm0.4}$ | $49.5_{\pm0.2}$ |
| CLIP* (16k) | $79.5_{\pm0.1}$ | $81.7_{\pm0.1}$ | $90.4_{\pm0.1}$ | $93.7_{\pm0.0}$ | $88.6_{\pm0.1}$ | $110.6_{\pm0.6}$ | $59.8_{\pm0.9}$ | $50.2_{\pm0.4}$ | $56.8_{\pm0.3}$ | $49.6_{\pm0.3}$ |

## B.2 Scaling properties

Fig. 8 and 9 show the performance of frozen Cap, CapPa, and CLIP* encoders on a variety of tasks as a function of the number of training examples seen and the encoder model size, respectively. Specifically, we evaluate our models on classification, captioning, and VQA when combined with a decoder trained from scratch to solve all those tasks jointly (following [2]), and on 10-shot linear classification based on the pre-logit features.

Tables 13, 14 and 15, 16 show the data from Fig. 8 and 9, respectively, in tabular form. For completeness, in Table 16 we also present the ImageNet zero-shot accuracy (without prompts) of CapPa and CLIP* models obtained with their respective pretrained decoder and encoder. We emphasize that scoring-based zero-shot classification is not the focus of this paper, and we did not optimize the Cap/CapPa models for this.

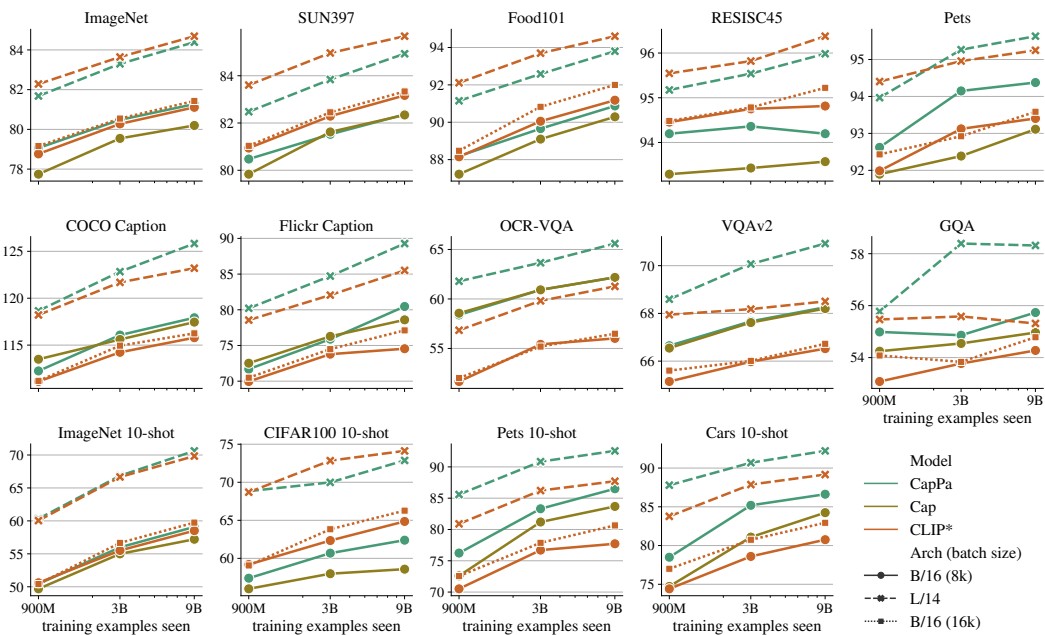

Figure 8: Performance of vision backbones pretrained with captioning (Cap/CapPa) and contrastive objective (CLIP*) as a function of the number of pretraining examples seen (expands the results in Fig. 2). **Top two rows:** Classification, captioning, and VQA performance with a decoder trained from scratch in multi-task fashion (see [2] for details). We use CIDEr for captioning, the VQAv2 weighted accuracy for VQAv2, and exact matching accuracy for all other tasks. **Bottom row:** 10-shot linear classification accuracy on the frozen pre-logit representation.

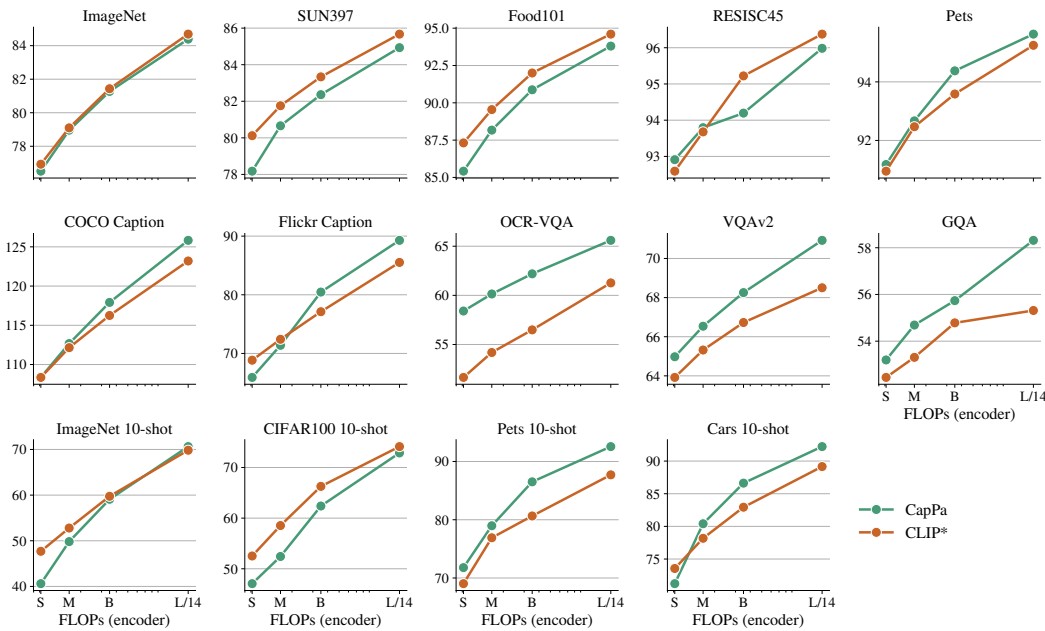

Figure 9: Performance of vision backbones pretrained with captioning (CapPa) and contrastive objective (CLIP*) as a function of the model size/FLOPs (we compare ViT-S/16, M/16, B/16, and L/14; this expands the results in Fig. 2). **Top two rows:** Classification, captioning, and VQA performance with a decoder trained from scratch in multi-task fashion (see [2] for details). We use CIDEr for captioning, the VQAv2 weighted accuracy for VQAv2, and exact matching accuracy for all other tasks. **Bottom row:** 10-shot linear classification accuracy on the frozen pre-logit representation.

Table 13: Data corresponding to Fig. 8 (top two rows) in tabular form. See caption of Fig. 8 for details on the metrics.

| | | | Classification | | | | | Captioning | | OCR | Question Ans. | |
|---|---|---|---|---|---|---|---|---|---|---|---|---|
| ex. seen | model | arch | i1k | sun | food | res | pet | COCO | Flickr | VQA | VQAv2 | GQA |
| 900M | Cap | B/16 (8k) | 77.7 | 79.8 | 87.2 | 93.3 | 91.9 | 113.5 | 72.5 | 58.6 | 66.5 | 54.2 |
| | CapPa | B/16 (8k) | 79.1 | 80.5 | 88.2 | 94.2 | 92.6 | 112.2 | 71.7 | 58.4 | 66.6 | 55.0 |
| | | L/14 | 81.7 | 82.5 | 91.1 | 95.2 | 94.0 | 118.7 | 80.2 | 61.8 | 68.6 | 55.8 |
| | CLIP* | B/16 (8k) | 78.8 | 80.9 | 88.2 | 94.5 | 92.0 | 111.1 | 70.0 | 51.7 | 65.1 | 53.1 |
| | | B/16 (16k) | 79.2 | 81.0 | 88.5 | 94.5 | 92.4 | 111.2 | 70.5 | 52.0 | 65.6 | 54.1 |
| | | L/14 | 82.3 | 83.6 | 92.1 | 95.5 | 94.4 | 118.2 | 78.6 | 56.8 | 67.9 | 55.5 |
| 3B | Cap | B/16 (8k) | 79.5 | 81.6 | 89.1 | 93.4 | 92.4 | 115.6 | 76.3 | 60.9 | 67.6 | 54.5 |
| | CapPa | B/16 (8k) | 80.5 | 81.5 | 89.7 | 94.4 | 94.1 | 116.1 | 75.9 | 60.9 | 67.7 | 54.9 |
| | | L/14 | 83.3 | 83.8 | 92.6 | 95.5 | 95.3 | 122.8 | 84.7 | 63.7 | 70.1 | 58.4 |
| | CLIP* | B/16 (8k) | 80.3 | 82.3 | 90.1 | 94.8 | 93.1 | 114.2 | 73.8 | 55.4 | 66.0 | 53.8 |
| | | B/16 (16k) | 80.5 | 82.5 | 90.8 | 94.8 | 92.9 | 114.9 | 74.5 | 55.2 | 66.0 | 53.8 |
| | | L/14 | 83.6 | 85.0 | 93.7 | 95.8 | 95.0 | 121.7 | 82.1 | 59.8 | 68.2 | 55.6 |
| 9B | Cap | B/16 (8k) | 80.2 | 82.3 | 90.3 | 93.6 | 93.1 | 117.5 | 78.6 | 62.2 | 68.2 | 55.0 |
| | CapPa | B/16 (8k) | 81.3 | 82.4 | 90.9 | 94.2 | 94.4 | 117.9 | 80.5 | 62.2 | 68.3 | 55.7 |
| | | L/14 | 84.4 | 84.9 | 93.8 | 96.0 | 95.6 | 125.8 | 89.3 | 65.6 | 70.9 | 58.3 |
| | CLIP* | B/16 (8k) | 81.1 | 83.2 | 91.2 | 94.8 | 93.4 | 115.8 | 74.5 | 56.0 | 66.5 | 54.3 |
| | | B/16 (16k) | 81.4 | 83.3 | 92.0 | 95.2 | 93.6 | 116.3 | 77.1 | 56.5 | 66.7 | 54.8 |
| | | L/14 | 84.7 | 85.7 | 94.6 | 96.4 | 95.2 | 123.2 | 85.5 | 61.3 | 68.5 | 55.3 |

Table 14: Data corresponding to Fig. 8 (bottom row) in tabular form. 10-shot linear classification accuracy on the frozen pre-logit features.

| ex. seen | model | arch | ImageNet | CIFAR100 | Pets | Cars |
|---|---|---|---|---|---|---|
| 900M | Cap | B/16 (8k) | 49.7 | 56.0 | 72.6 | 74.7 |
| | CapPa | B/16 (8k) | 50.4 | 57.4 | 76.2 | 78.5 |
| | | L/14 | 60.3 | 68.8 | 85.6 | 87.8 |
| | CLIP* | B/16 (8k) | 50.6 | 59.2 | 70.5 | 74.4 |
| | | B/16 (16k) | 50.4 | 59.1 | 72.6 | 77.0 |
| | | L/14 | 60.0 | 68.7 | 80.9 | 83.8 |
| 3B | Cap | B/16 (8k) | 55.0 | 58.0 | 81.2 | 81.1 |
| | CapPa | B/16 (8k) | 56.0 | 60.7 | 83.3 | 85.2 |
| | | L/14 | 66.9 | 70.0 | 90.8 | 90.7 |
| | CLIP* | B/16 (8k) | 55.5 | 62.3 | 76.7 | 78.6 |
| | | B/16 (16k) | 56.7 | 63.8 | 77.9 | 80.7 |
| | | L/14 | 66.7 | 72.8 | 86.2 | 87.9 |
| 9B | Cap | B/16 (8k) | 57.2 | 58.6 | 83.7 | 84.2 |
| | CapPa | B/16 (8k) | 59.1 | 62.4 | 86.5 | 86.6 |
| | | L/14 | 70.6 | 72.9 | 92.6 | 92.2 |
| | CLIP* | B/16 (8k) | 58.5 | 64.9 | 77.7 | 80.8 |
| | | B/16 (16k) | 59.7 | 66.3 | 80.6 | 82.9 |
| | | L/14 | 69.8 | 74.1 | 87.7 | 89.2 |

Table 15: Data corresponding to Fig. 9 (top two rows) in tabular form. See caption of Fig. 9 for details on the metrics

| arch | FLOPs | model | Classification | | | | | Captioning | | OCR | Question Ans. | |
|---|---|---|---|---|---|---|---|---|---|---|---|---|
| | | | i1k | sun | food | res | pet | COCO | Flickr | VQA | VQAv2 | GQA |
| S/16 | 9.2G | CapPa | 76.5 | 78.2 | 85.4 | 92.9 | 91.2 | 108.4 | 65.9 | 58.4 | 65.0 | 53.2 |
| | | CLIP* | 76.9 | 80.1 | 87.3 | 92.6 | 91.0 | 108.4 | 68.8 | 51.7 | 63.9 | 52.4 |
| M/16 | 16.0G | CapPa | 79.0 | 80.7 | 88.2 | 93.8 | 92.7 | 112.7 | 71.4 | 60.1 | 66.5 | 54.7 |
| | | CLIP* | 79.1 | 81.8 | 89.5 | 93.7 | 92.5 | 112.2 | 72.4 | 54.2 | 65.3 | 53.3 |
| B/16 | 35.1G | CapPa | 81.3 | 82.4 | 90.9 | 94.2 | 94.4 | 117.9 | 80.5 | 62.2 | 68.3 | 55.7 |
| | | CLIP* | 81.4 | 83.3 | 92.0 | 95.2 | 93.6 | 116.3 | 77.1 | 56.5 | 66.7 | 54.8 |
| L/14 | 161.8G | CapPa | 84.4 | 84.9 | 93.8 | 96.0 | 95.6 | 125.8 | 89.3 | 65.6 | 70.9 | 58.3 |
| | | CLIP* | 84.7 | 85.7 | 94.6 | 96.4 | 95.2 | 123.2 | 85.5 | 61.3 | 68.5 | 55.3 |

Table 16: Data corresponding to Fig. 9 (bottom row) in tabular form. 10-shot linear classification accuracy on the frozen pre-logit features. We also show the ImageNet zero-shot classification accuracy (without prompts) when using the pretrained text encoder (CLIP*) or text decoder with scoring (CapPa) for reference (last column).

| arch | FLOPs | model | ImageNet | CIFAR100 | Pets | Cars | ImageNet zs. |
|---|---|---|---|---|---|---|---|
| S/16 | 9.2G | CapPa | 40.6 | 47.1 | 71.8 | 71.2 | 35.1 |
| | | CLIP* | 47.7 | 52.5 | 69.0 | 73.6 | 52.8 |
| M/16 | 16.0G | CapPa | 49.8 | 52.4 | 79.0 | 80.4 | 43.0 |
| | | CLIP* | 52.8 | 58.5 | 76.9 | 78.2 | 58.7 |
| B/16 | 35.1G | CapPa | 59.1 | 62.4 | 86.5 | 86.6 | 52.7 |
| | | CLIP* | 59.7 | 66.3 | 80.6 | 82.9 | 64.1 |
| L/14 | 161.8G | CapPa | 70.6 | 72.9 | 92.6 | 92.2 | 63.8 |
| | | CLIP* | 69.8 | 74.1 | 87.7 | 89.2 | 71.2 |

## B.3 Attribution, relation, ordering

Table 17 shows extended results for different models on the ARO benchmark [67] (see Table 6 in the main paper). In addition to the clear superiority of Cap/CapPa over CLIP* models discussed in the main paper, it can be observed that increasing the model capacity form B/16 to L/14 leads to an overall improvement for CapPa, while this is not the case for CLIP*.

Table 17: Results on the Attribute, Relation and Order (ARO) benchmark [67]. Cap and CapPa models clearly outperform all CLIP* and CLIP variants across all data sets, even when training the model to be sensitive to word ordering and attribution as in NegCLIP [67]. Values for "ARO Best" are taken from [67]. "Blind dec." corresponds to Cap without vision encoder, i.e. the vision encoder features fed to the decoder are replaced with all zeros.

| | Arch | VG Attribution | VG Relation | Flickr Order | COCO Order |
|---|---|---|---|---|---|
| Blind dec. | - | 83.7 | 86.2 | 98.8 | 98.7 |
| Cap | B/16 | 88.9 | 86.6 | 99.1 | 99.0 |
| CapPa | B/16 | 85.7 | 86.7 | 99.2 | 98.8 |
| CapPa | L/14 | 89.3 | 86.0 | 99.3 | 99.0 |
| CLIP* (8k) | B/16 | 55.4 | 39.8 | 43.7 | 32.8 |
| CLIP* (16k) | B/16 | 53.2 | 39.7 | 45.5 | 37.0 |
| CLIP* (16k) | L/14 | 57.8 | 35.9 | 40.2 | 31.5 |
| CLIP | B/32 | 63.2 | 59.1 | 59.4 | 47.3 |
| CLIP | B/16 | 62.7 | 58.7 | 57.9 | 49.5 |
| ARO Best | - | 88.0 | 73.0 | 60.0 | 46.0 |
| NegCLIP | B/32 | 71.0 | 81.0 | 91.0 | 86.0 |

## B.4 SugarCrepe

We provide the full breakdown of results across all our models on SugarCrepe in Table 18. The numbers for OpenCLIP are taken from [21] and represent the best, largest contrastive model that was benchmarked on SugarCrepe to date. Even the small ViT-B/16 Cap model significantly outperforms it on all but the "Replace Object" task, which is a task that matches contrastive's "bag of word"-style of learning well.

Table 18: Full results on the SugarCrepe [21] benchmark suite.

| Training | Arch | Replace | | | Swap | | Add | |
|---|---|---|---|---|---|---|---|---|
| | | Object | Attribute | Relation | Object | Attribute | Object | Attribute |
| Cap | B/16 | 91.10 | 88.32 | 85.21 | 79.27 | **88.74** | 98.59 | 99.28 |
| CapPa | B/16 | 89.95 | 88.71 | 84.35 | 80.49 | 85.74 | 98.84 | **99.42** |
| CapPa | L/14 | 92.01 | **90.10** | **87.34** | **82.11** | 88.44 | **98.93** | **99.42** |
| CLIP* (8k) | B/16 | 93.70 | 82.36 | 66.29 | 61.79 | 67.12 | 83.46 | 76.01 |
| CLIP* (16k) | B/16 | 94.07 | 84.64 | 67.14 | 60.98 | 65.47 | 86.37 | 77.46 |
| CLIP* (16k) | L/14 | 95.70 | 84.26 | 69.06 | 65.04 | 68.02 | 86.76 | 78.32 |
| OpenCLIP | G/14 | **96.67** | 88.07 | 74.75 | 62.20 | 74.92 | 92.19 | 84.54 |

We further show qualitative examples in Tables 22–24. The examples are manually picked to be representative (we show wins and losses), while avoiding uninteresting (i.e. seemingly random), too cluttered, or too verbose examples. Thus, the examples are cherry-picked to be presentable, but are meant to be representative. All images are from the COCO validation set.

Each image comes with a positive and a (hard) negative caption, and a model's prediction is deemed correct when it scores the positive caption higher than the negative one. For the CapPa model, we score each caption using the log-likelihood, meaning negative numbers closer to zero correspond to a higher score (i.e. a score of -20 means the caption fits the image more than a score of -110). For the CLIP model, we score each caption using the dot-product of normalized embedding similarity as is usual, but we multiply the resulting score by 100 for readability.

Table 19: Impact of decoder architecture design choices in Cap on 10-shot linear classification accuracy: **Left:** Effect of sharing the embedding between decoder input and output and removing biases from decoder layers. **Right:** Effect of the number of decoder layers.

| share emb. | dec. bias | ImageNet | CIFAR100 | Pets | Cars |
|---|---|---|---|---|---|
| yes | no | 47.8 | 55.8 | 71.5 | 71.7 |
| no | no | 49.7 | 56.0 | 72.6 | 74.7 |
| yes | yes | 48.3 | 54.6 | 74.4 | 70.2 |
| no | yes | 49.3 | 56.6 | 72.7 | 71.9 |

| dec. layers | ImageNet | CIFAR100 | Pets | Cars |
|---|---|---|---|---|
| 3 | 48.7 | 53.7 | 73.5 | 73.7 |
| 6 | 49.7 | 56.0 | 72.6 | 74.7 |
| 12 | 48.7 | 54.8 | 74.4 | 73.8 |

We noticed that for the **Add** scenarios, where CapPa performs almost perfectly, the only losses are due to typos in the positive caption ("toliet' instead of "toilet" and "bridge" instead of "bride"), so we also provide the score for the corrected caption in the *Pos (fixed)*, which confirms the typos are the reason for the model failure.

### B.5 Ablations: Decoder architecture

While following the original transformer decoder architecture [60] closely, we investigate several modifications that have become common in the literature [52, 11]. Specifically, we ablate the effect of removing biases in decoder layers, as well as sharing the decoder input and output embeddings. Table 19 (left) shows that not sharing the embeddings leads to overall better 10-shot accuracy than sharing them, and additionally removing the decoder biases does not hurt. Furthermore, we observed significantly improved stability across encoder architectures, scales and training schedules when removing the decoder biases.

Table 19 (right) reveals that the overall best 10-shot classification accuracy is obtained when using a 6 layer decoder. This decoder depth also leads to a total parameter count comparable to the corresponding CLIP* model (Table 1).

### B.6 Further Ablations

Table 20 compares the performance of the CapPa and CLIP* vision encoders with a ViT-B/16 pretrained in supervised fashion on ImageNet-21k when combined with transformer decoder.

Table 21 represents the data from Fig. 6 in tabular form.

Table 20: Comparison of CapPa and CLIP* with a ViT-B/16 pretrained in supervised fashion on ImageNet-21k (we use the checkpoint from Steiner et al. 2021) when combined with a transformer decoder trained from scratch for classification, captioning, and VQA [2]. CLIP* and CapPa outperform the model pretrained in supervised fashion.

| | Classification | | | | | Captioning | | OCR | Question Ans. | |
|---|---|---|---|---|---|---|---|---|---|---|
| | i1k | sun | food | res | pet | COCO | Flickr | VQA | VQAv2 | GQA |
| ViT-B/16 (i21k) | $73.1_{\pm0.1}$ | $72.8_{\pm0.2}$ | $81.2_{\pm0.2}$ | $86.2_{\pm0.1}$ | $85.6_{\pm0.3}$ | $95.0_{\pm0.4}$ | $52.3_{\pm0.1}$ | $39.1_{\pm0.1}$ | $57.6_{\pm0.3}$ | $50.1_{\pm0.1}$ |
| CapPa | $81.3_{\pm0.1}$ | $82.4_{\pm0.1}$ | $90.9_{\pm0.1}$ | $94.2_{\pm0.2}$ | $94.4_{\pm0.1}$ | $117.9_{\pm0.6}$ | $80.5_{\pm0.2}$ | $62.2_{\pm0.0}$ | $68.3_{\pm0.1}$ | $55.7_{\pm0.2}$ |
| CLIP* | $81.4_{\pm0.1}$ | $83.3_{\pm0.1}$ | $92.0_{\pm0.1}$ | $95.2_{\pm0.2}$ | $93.6_{\pm0.2}$ | $116.3_{\pm0.7}$ | $77.1_{\pm0.7}$ | $56.5_{\pm0.1}$ | $66.7_{\pm0.1}$ | $54.8_{\pm0.6}$ |

## C Societal impact

Our models fit in the broader context of large scale vision-language pretraining and as such share many of the benefits and issues of related models such as [50, 26, 66, 40, 63]: They produce versatile vision models which obtain strong performance on natural images, on OCR-related tasks, and also when combined with a generative language decoder. These capabilities enable many useful applications (e.g. assistive technologies, medical imaging), but also potentially harmful ones (e.g. surveillance). We generally recommend either employing the CapPa vision encoder with a new, task-specific prediction head, or using the pretrained decoder for scoring only. We do not recommend the pretrained decoder for downstream image captioning applications without further refinement, as

Table 21: Ablation results representing Fig. 6 in tabular form. **Left:** 10-shot linear classification accuracy based on the frozen encoder representation as a function of the fraction of training batches for which parallel prediction is performed in CapPa. **Right:** 10-shot linear classification accuracy and zero-shot classification accuracy as a function of the vision encoder architecture.

| fraction | INet | C100 | Pets | Cars |
|---|---|---|---|---|
| 0% | 49.7 | 56.0 | 72.6 | 74.7 |
| 25% | 46.7 | 52.9 | 71.9 | 72.8 |
| 50% | 49.8 | 57.8 | 76.8 | 76.9 |
| 75% | 50.4 | 57.4 | 76.2 | 78.5 |
| 90% | 49.0 | 59.5 | 73.1 | 79.0 |

| arch | model | INet | C100 | Pets | Cars | INet zs. |
|---|---|---|---|---|---|---|
| R50 | CLIP* (8k) | 39.8 | 33.5 | 49.2 | 60.9 | 43.6 |
| | Cap | 37.8 | 33.3 | 48.6 | 52.4 | 28.5 |
| ViT-B/32 | CLIP* (8k) | 44.1 | 57.7 | 64.7 | 68.1 | 48.3 |
| | Cap | 41.0 | 53.7 | 64.0 | 58.7 | 35.4 |
| ViT-B/16 | CLIP* (8k) | 50.6 | 59.2 | 70.5 | 74.4 | 52.2 |
| | Cap | 49.7 | 56.0 | 72.6 | 74.7 | 43.8 |

it is trained on a large number of alt-texts from the web. Harmful biases should be carefully assessed in the context of the concrete downstream application and prediction head used. For example, when combining the encoder with a (potentially pretrained) decoder for captioning or VQA, an assessment of hallucinations, attribute binding issues and stereotypical attribution should be done.

Table 22: Representative examples of CapPa L/14 wins (first three) and losses (fourth) over CLIP L/14 on the different **replace** categories of the SugarCrepe hard negatives benchmark suite. See text for example selection criteria. Higher score means better: for CapPa this is the log-likelihood, so closer to 0 is better, while for CLIP this is the matching score (multiplied by 100) so closer to 100 is better.

| | | CapPa wins over CLIP | | | CLIP wins over CapPa |
|---|---|---|---|---|---|
| **Replace Object** | Positive | CapPa:**-28.6** CLIP:**23.6** *Street signs on the corner of Gladys and Detroit* | CapPa:**-45.4** CLIP:**13.2** *A run down building with two planters outside the door* | CapPa:**-42.4** CLIP:**16.8** *A brown bird has a small yellow head.* | CapPa:**-54.6** CLIP:**13.7** *The model toys are positioned on the table.* |
| | Negative | CapPa:**-53.8** CLIP:**13.9** *Street signs on the corner of Gladys and Chicago.* | CapPa:**-59.2** CLIP:**14.8** *A run down building with a statue outside the door.* | CapPa:**-46.2** CLIP:**18.1** *A brown bird has a small yellow beak.* | CapPa:**-51.7** CLIP:**8.8** *The books are positioned on the table.* |

| | | | | | |
|---|---|---|---|---|---|
| **Replace Attribute** | Positive | CapPa:**-45.0** CLIP:**14.7** *A plate of food with a fried egg and colorful vegetables.* | CapPa:**-47.3** CLIP:**13.1** *A bunch of different foods on display on a counter.* | CapPa:**-17.3** CLIP:**15.7** *A large black truck in a parking lot.* | CapPa:**-115.9** CLIP:**14.1** *Two large trucks are travelling along a tree-lined roadway.* |
| | Negative | CapPa:**-59.5** CLIP:**15.1** *A plate of food with a fried egg and monochromatic vegetables.* | CapPa:**-53.8** CLIP:**14.8** *A bunch of similar foods on display on a counter.* | CapPa:**-38.3** CLIP:**16.1** *A small black truck in a parking lot.* | CapPa:**-61.7** CLIP:**12.5** *Two large trucks are travelling along a deserted roadway.* |

| | | | | | |
|---|---|---|---|---|---|
| **Replace Relation** | Positive | CapPa:**-20.0** CLIP:**18.5** *A fire hydrant in a grassy field next to a bush* | CapPa:**-48.2** CLIP:**18.6** *A cell phone on top of a calculator near a computer keyboard.* | CapPa:**-29.1** CLIP:**24.4** *A red fire hydrant on a city sidewalk.* | CapPa:**-55.6** CLIP:**17.6** *A train driving over a small bridge on a green hillside.* |
| | Negative | CapPa:**-56.1** CLIP:**21.5** *A fire hydrant in a grassy field far from a bush.* | CapPa:**-56.1** CLIP:**19.2** *A cell phone underneath a calculator near a computer keyboard.* | CapPa:**-35.6** CLIP:**25.6** *A red fire hydrant beside a city sidewalk.* | CapPa:**-54.7** CLIP:**17.4** *A train passing under a small bridge on a green hillside.* |

Table 23: Representative examples of CapPa L/14 wins (first three) and losses (fourth) over CLIP L/14 on the different **add** categories of the SugarCrepe hard negatives benchmark suite. See text for example selection criteria. Higher score means better: for CapPa this is the log-likelihood, so closer to 0 is better, while for CLIP this is the matching score (multiplied by 100) so closer to 100 is better.

| | | CapPa wins over CLIP | | | CLIP wins over CapPa |
|---|---|---|---|---|---|
| **Add Object** | Positive | CapPa:**-18.2** CLIP:**13.7** *A bathroom with a mirror and a sink.* | CapPa:**-30.2** CLIP:**14.3** *A two layered cake sits on a table top* | CapPa:**-27.3** CLIP:**14.0** *an image of a plate of food with meat and veggies* | CapPa:**-60.3** CLIP:**22.5** *A bridge and groom cutting their wedding cake that has fruit on top.* |
| | Negative | CapPa:**-150.3** CLIP:**13.7** *A bathroom with a mirror, sink, and shower.* | CapPa:**-64.9** CLIP:**15.5** *A two layered cake sits on a table top next to a vase of flowers.* | CapPa:**-148.6** CLIP:**14.6** *An image of a plate of food with meat, fruit, and veggies.* | CapPa:**-53.4** CLIP:**21.6** *A bride and groom cutting their wedding cake that has flowers and fruit on top.* |
| | |  |  |  |  |
| | Pos (fixed) | | | | CapPa:**-46.1** CLIP:**22.7** *A bride and groom cutting their wedding cake that has fruit on top.* |
| **Add Attribute** | Positive | CapPa:**-43.8** CLIP:**21.0** *A little girl smiling for the camera with an umbrella behind her.* | CapPa:**-65.4** CLIP:**16.1** *A clock fastened to a brick store front reads 10 after 10* | CapPa:**-49.9** CLIP:**17.6** *A person frying some kind of food on a stove.* | CapPa:**-62.1** CLIP:**20.4** *There is a stuffed animal sitting on the toliet.* |
| | Negative | CapPa:**-121.5** CLIP:**21.0** *A little girl smiling for the camera with a polka-dotted umbrella behind her.* | CapPa:**-90.7** CLIP:**17.0** *A clock fastened to a lush brick store front reads 10 after 10.* | CapPa:**-115.3** CLIP:**19.5** *A person frying some curry-spiced food on a stove.* | CapPa:**-49.8** CLIP:**19.1** *There is a stuffed animal sitting on the decorated toilet.* |
| | |  |  |  |  |
| | Pos (fixed) | | | | CapPa:**-36.8** CLIP:**21.3** *There is a stuffed animal sitting on the toilet.* |

Table 24: Representative examples of CapPa L/14 wins (first three) and losses (fourth) over CLIP L/14 on the different **swap** categories of the SugarCrepe hard negatives benchmark suite. See text for example selection criteria. Higher score means better: for CapPa this is the log-likelihood, so closer to 0 is better, while for CLIP this is the matching score (multiplied by 100) so closer to 100 is better.

| | | CapPa wins over CLIP | | | CLIP wins over CapPa |
|---|---|---|---|---|---|
| **Swap Object** | Positive | CapPa:**-33.5** CLIP:**22.5** *a bright kitchen with tulips on the table and plants by the window* | CapPa:**-54.9** CLIP:**22.1** *A person cutting a pizza next to a salad and bottles of wine on wooden table.* | CapPa:**-38.1** CLIP:**15.6** *A close up of a sandwich with a drink in the back.* | CapPa:**-111.4** CLIP:**20.5** *Statues on the second floor of a building, sitting below a clock.* |
| | Negative | CapPa:**-56.8** CLIP:**22.8** *A bright kitchen with plants on the table and tulips by the window.* | CapPa:**-57.5** CLIP:**22.5** *A person cutting a salad next to a pizza and bottles of wine on wooden table.* | CapPa:**-45.6** CLIP:**16.2** *A close up of a drink with a sandwich in the back.* | CapPa:**-110.8** CLIP:**19.4** *A clock on the second floor of a building, sitting below statues.* |

| | | CapPa wins over CLIP | | | CLIP wins over CapPa |
|---|---|---|---|---|---|
| **Swap Attribute** | Positive | CapPa:**-32.6** CLIP:**15.4** *a white cake is by a bunch of flowers* | CapPa:**-45.9** CLIP:**19.4** *A blue tennis racket has a yellow tennis ball on it.* | CapPa:**-28.4** CLIP:**16.3** *a black bike rests against a brown bed* | CapPa:**-108.6** CLIP:**19.3** *All of the cows are poking their heads out, eating some hay.* |
| | Negative | CapPa:**-64.1** CLIP:**17.3** *A bunch of cakes are by a white flower.* | CapPa:**-54.9** CLIP:**19.5** *A yellow tennis racket has a blue tennis ball on it.* | CapPa:**-52.8** CLIP:**16.9** *a brown bike rests against a black bed.* | CapPa:**-107.1** CLIP:**18.2** *Some cows are poking their heads out, eating all of the hay.* |

