# OpenReview forum: "Image Captioners Are Scalable Vision Learners Too"
_NeurIPS.cc/2023/Conference — NeurIPS 2023 oral_

### Official Review · Reviewer_mqQw · 2023-06-13

**Soundness:** 4 excellent
**Presentation:** 4 excellent
**Contribution:** 3 good
**Rating:** 8
**Confidence:** 5

**Summary:**

This paper shows that training a ViT with the image-to-text generation (i.e., image captioning) objective is an effective way to learn good visual representations for downstream tasks. The paper systematically compares between the image captioning and the image-text contrastive learning (CLIP) objectives, where image captioning leads to ViTs that are comparable to the CLIP-ViT. Furthermore, the paper proposes parallel decoding as an additional pre-training objective to complement the conventional autoregressive decoding.

**Strengths:**

- The paper systematically demonstrates a refreshing observation: image-to-text generative learning leads to vision encoders as good as contrastive-learned ones. This opens up more research opportunities for visual representation learning using language supervision.

- The pre-training details are well-controlled to ensure a fair comparison between Cap and CLIP.

- The paper performs a comprehensive evaluation on the pre-trained ViT, including both image classification tasks and vision-language tasks. Table 8 is very nice in particular to compare the frozen ViTs.

- It is interesting to see that Cap/CapPa outperforms CLIP on tasks that require fine-grained language understanding.

- The proposed parallel prediction makes sense intuitively as a way to enforce stronger supervision on the ViT.


**Weaknesses:**

I do not find major weakness from this paper. There is a minor limitation as detailed below. More questions about the paper are listed in the next section.

Minor limitation: Since Cap-ViT is pre-trained using an encoder-decoder paradigm, its representations would be more suited for similar encoder-decoder tasks (image caption, VQA) compared to CLIP-ViT. Therefore, frozen adaption may not justify the advantage of Cap-ViT on such tasks. It would be good to also report fine-tuning performance.


**Questions:**

- How does the size of the text decoder affect the representation learning performance?

- What if the pre-training is performed using a pre-trained text decoder such as T5, does it improve representation learning?

- Would Cap and CLIP have a complementary effect if they are combined as a multi-task pre-training objective such as in BLIP?

**Limitations:**

Yes the authors have addressed the limitations.

---

> ### Author Rebuttal · Authors · 2023-08-09
>
> **Minor limitation: Frozen adaption may not justify the advantage of Cap-ViT**
>
> We actually tried fine-tuning the image encoder along with the decoder for both CLIP* and Cap/CapPa models and did not obtain an improvement for any of the models. This is consistent with prior work which did not observe an improvement either for CLIP-style models when fine-tuning with the same decoder-based setup, see [2, Sec.5.7]. Frozen adaptation is also favorable from a computational perspective as no gradients have to be propagated through the encoder.
>
> \
> **How does the size of the text decoder affect the representation learning performance?**
>
> An ablation of the decoder depth for Cap can be found in Table 7 (right). 3 and 12 layer decoders obtain a 1-3 points lower 10-shot classification accuracy across the majority of the eval sets compared with the (default) 6 layer decoder.
>
> \
> **What if the pre-training is performed using a pre-trained text decoder such as T5, does it improve representation learning?**
>
> This is an interesting question. We investigated this setup with a pretrained T5-Base decoder (which has 12 instead of 6 decoder layers as Cap/CapPa). First, to enable stable training we needed to reduce the learning rate from 1e-3 to 5e-4 and set the optimizer beta2 parameter to 0.95. Furthermore, training was only stable when unfreezing the cross-attention parameters. We also explored variants where we re-initialized the cross-attention parameters, and another variant where additionally all decoder parameters were trained. For all other design choices we follow the Cap-ViT B/16 setup with the 900M example schedule. None of these variants performs better than Cap overall, and the more we allow the decoder to deviate from its language-pretrained weights the better the vision performance gets.
>
> Using a pretrained T5 decoder (10-shot linear eval.)
> | model                          |   ImageNet |   CIFAR100 |   Pets |   Cars |
> |:-------------------------------|-----------:|-----------:|-------:|-------:|
> | Cap                            |       49.7 |       56.0 |   72.6 |   74.7 |
> | Cap (12 dec. layers)           |       48.7 |       54.8 |   74.4 |   73.8 |
> | Cap T5                         |       42.8 |       44.9 |   69.3 |   62.3 |
> | Cap T5 (rein. xatt.)           |       43.7 |       45.7 |   68.3 |   66.9 |
> | Cap T5 (rein. xatt., unfreeze) |       48.6 |       55.2 |   72.0 |   75.6 |
>
> \
> **Would Cap and CLIP have a complementary effect if they are combined as a multi-task pre-training objective such as in BLIP?**
>
> While there is anecdotal evidence in the literature that these losses can have complementary effects (see e.g. [32, 33, 52]) these experiments do not take pretraining compute into account and/or do not remove unnecessary network components when ablating losses. By contrast, our experiments control for these factors and show that both Cap/CapPa and CLIP lead to competitive vision encoders. It is therefore unclear how much complementary signal they provide when controlling for model capacity and pretraining compute. We leave analysis of this aspect at scale for future work.

---

> > ### Comment · Reviewer_mqQw · 2023-08-11
> >
> > I appreciate the authors' response and the additional experiments. I confirm my original score of strong accept. This paper provides interesting and refreshing observations with strong experimental support.

---

### Official Review · Reviewer_9JDL · 2023-06-24

**Soundness:** 4 excellent
**Presentation:** 2 fair
**Contribution:** 3 good
**Rating:** 6
**Confidence:** 4

**Summary:**

This paper shows that the captioning loss is a competitive alternative to pretrain image backbones compared with contrastive loss (CLIP) when using the same training budget and training data. The standard next token prediction objective on the full caption sequence (Cap) is complemented with a parallel prediction loss (CapPa) which is used on a quarter of the training batches. Experiments notably show:
- the benefit of CapPa over Cap in a variety of settings
- for classification, CLIP models are better than CapPa models at linear probing but the gap is bridged when using MAP probing.
- with LiT transfer, CapPa models perform competitively in classification and are better at vision and language tasks like VQA or image captioning.
- CapPa models scale well with model size and training budget.
- CapPa models are better at attribute / relation / order prediction.


**Strengths:**

- Alternating parallel prediction with autoregressive prediction improves downstream transfer.
- The experimental setup is well described.
- Extensive experiments with interesting insights, e.g., the CapPa backbones are most competitive when exploiting all the token embeddings (and not merely average pooling them) which makes sense as they are all used for cross-attention.


**Weaknesses:**

- The layout of the tables and figures is hard to follow: Table 2 appears before Table 1, Table 5 is not discussed anywhere, Figure 3 is placed long after it is discussed, and Table 10 (L238) does not exist. Explaining the CLIP* and 8k/16k meaning in the caption of Table 2 would also help.
- How do the compute requirements (e.g. gpu memory) of the CLIP* with 8k/16k batch size compare with CapPa?
- Most experiments use a proprietary dataset (WebLi) for pretraining and the code is not provided, which harms reproducibility.


**Questions:**

See weaknesses.

**Limitations:**

Limitations are discussed in Section 5. Potential negative societal impact is not discussed.

---

> ### Author Rebuttal · Authors · 2023-08-09
>
> **The layout of the tables and figures is hard to follow**
>
> Thank you for bringing this to our attention. We will improve the alignment of tables and figures with the text flow in the next revision.
>
> \
> **How do the compute requirements (e.g. gpu memory) of the CLIP\* with 8k/16k batch size compare with CapPa?**
>
> The default Cap/CapPa model and training setup is chosen such that it closely matches the corresponding CLIP* setup in terms of actual accelerator hours (and examples seen). Table 1 in the paper reports TPUv4 hours per billion examples seen along with parameter count (which is also matched for the two model families). Below are the memory requirements per chip when training the models on 64 TPUv4 chips (the same setup as used to determine TPU hours in Table 1). The memory requirements of CapPa and CLIP* are comparable for the same batch size as well.
>
> | model			| TPUv4 mem |
> |:------------------|-----------:|
> | Cap/CapPa B/16	| 19.62 GiB |
> | CLIP* (8k) B/16	| 20.05 GiB |
> | CLIP* (16k) B/16	| 31.46 GiB |
>
> \
> **Most experiments use a proprietary dataset (WebLi) for pretraining and the code is not provided, which harms reproducibility.**
>
> The experiments on the publicly available LAION-400M data set (Sec. 4.4, Fig. 6) show that our most important results transfer to publicly available data. Furthermore, we only perform very simple text-based filtering of the WebLI data (which is from the public web) as done by prior work [4, 25], and we do not use any image-text similarity based filtering or other sophisticated filtering procedures. We are working on a code release and are hopeful to publish code before the conference.
>
> \
> **Potential negative societal impact is not discussed.**
>
> We plan to include the following discussion in the paper:
>
> Our models fit in the broader context of large scale vision-language pretraining and as such share many of the benefits and issues of related models such as [40, 21, 52, 33, 50]: They produce versatile vision models which obtain strong performance on natural images, on OCR-related tasks, and also when combined with a generative language decoder. These capabilities enable many useful applications (e.g. assistive technologies, medical imaging), but also potentially harmful ones (e.g. surveillance).
> We generally recommend either employing the CapPa vision encoder with a new, task-specific prediction head, or using the pretrained decoder for scoring only. We do not recommend the pretrained decoder for downstream image captioning applications without further refinement, as it is trained on a large number of alt-texts from the web. Harmful biases should be carefully assessed in the context of the concrete downstream application and prediction head used. For example, when combining the encoder with a (potentially pretrained) decoder for captioning or VQA, an assessment of hallucinations, attribute binding issues and stereotypical attribution should be done.

---

> > ### Comment · Reviewer_9JDL · 2023-08-10
> >
> > I thank the authors for providing a rebuttal, have read other reviews, and confirm that I am inclined to accept this paper. In particular, it is valuable to see that CapPa has similar memory requirements with CLIP* (8k). I encourage the authors to improve the layout of tables and figures and include this discussion of potential negative societal impact in the paper, and to release the code to improve reproducibility,

---

### Official Review · Reviewer_G8WT · 2023-07-06

**Soundness:** 4 excellent
**Presentation:** 4 excellent
**Contribution:** 4 excellent
**Rating:** 9
**Confidence:** 5

**Summary:**

This paper revisits image captioning as a pretraining task for learning general vision encoders from web image-text pairs. Surprisingly, the empirical study shows that captioning pre-trained vision encoders is competitive or better than contrastively pre-trained ones on image recognition and vision-language tasks.

**Strengths:**

- The paper presents a novel and interesting finding: using captioning as a pre-training scheme can also achieve strong results compared with contrastive ones. The empirical study is valuable to the community.
- The paper presents an in-depth analysis of different design factors, such as the use of decoders, encoders, and pre-training data. One can find many insightful discussions in the experiment section.


**Weaknesses:**

- Although captioning can be a promising scheme for pre-training, it may not be able to replace the existing contrastive pre-trained objective. For many established tasks, such as image-text retrieval or estimating the similarity between a given image-text pair, CLIP-style models are convenient and offer a more efficient computation.

**Questions:**

The paper is well-written and technically flawless. I don't have significant concerns about this paper.

**Limitations:**

No potential negative societal impact.

---

> ### Author Rebuttal · Authors · 2023-08-09
>
> **CLIP-style models are convenient for image-text retrieval/computing image-text similarities and offer a more efficient computation.**
>
> This is indeed a downside of captioning-based approaches. On the upside, Cap and CapPa are much better than CLIP-style models in zero-shot classification tasks where word order, attribution, and relation are important (see Table 6). Furthermore, we show that LiT-tuning [56], where a text encoder is trained to match a frozen image embedding, represents an efficient way to equip Cap/CapPa with these capabilities (Table 4).

---

> ### Comment · Reviewer_G8WT · 2023-08-20
>
> Thanks. I don't have further comments. I will keep my rating.

---

### Official Review · Reviewer_J9Jb · 2023-07-09

**Soundness:** 4 excellent
**Presentation:** 4 excellent
**Contribution:** 3 good
**Rating:** 8
**Confidence:** 5

**Summary:**

In this paper, the authors more a controlled comparison between two pretraining approaches to learning visual representations from language supervision:
contrastive (CLIP-style) and generative (= image captioning, e.g. VirTex, SimVLM, BLIP, CoCa), etc.
The goal of this paper is not to advance the state-of-the-art, but rather to observe the model/data scaling behavior of these pretraining tasks.
Experiments show that captioning-pretrained models can match or outperform CLIP-style contrastive models on several multi-modal tasks.


**Strengths:**

**I think this paper, in its current form, already matches the quality of a typical publication at the NeurIPS conference.**
I strongly recommend acceptance; it is relevant to the conference audience and will spur interesting discussion in the community.
Below I highlight the main strengths of the paper to substantially support my assessment:

1. **Paper presents contrary evidence to existing results:**
The vision community is making rapid progress with vision-language models as they enable new transfer applications that can be specified using natural language.
Much of this progress in the last 2-3 years was catalyzed by the development of CLIP (and concurrent works like ALIGN).
Ever since, the vision community has largely gravitated towards pushing progress on multi-modal contrastive models,
following the "image captioning models converge slower on web data" result from the CLIP paper.
This paper presents a piece of contrary evidence that simple captioning-only models can match or outperform their contrastive counterparts.

2. **Promising alternative to poor language understanding of contrastive models:**
Image captioning as a pretraining task has been studied in previous models (e.g. VirTex, SimVLM, BLIP, CoCa, etc.).
The main novelty of this work is a direct comparison of captioning with the contrastive objective at scale,
with controlled model capacity and training dataset size.
Hence, I think this is a timely contribution that will force practitioners and researchers to rethink the relevance of text-generative models
and side-step the embarrassing failures of contrastive models, e.g. their inability to distinguish "man eating a sandwich" from "sandwich eating a man".

3. **New evaluations with captioning models show practical runtime solutions for classification/retrieval:**
Captioning models are known for their slow image/text retrieval runtime since they cannot "cache" the text classifier weights once like contrastive models.
However, they are better at language understanding than contrastive models which are known to behave as bag-of-words models.
Authors show evaluations related to "LiT tuning" to convert a captioning-pretrained image encoder to a contrastive model --
this helps overcome the runtime overhead of image captioning-only models.

4. **Experiments are thorough and well presented:**
The paper studies a targeted comparison between captioning and contrastive pretraining approaches.
The authors present a series of experiments and evaluations to support this study.
All comparisons seem fair and controlled to the best of my knowledge, with differences specified wherever relevant.
Modeling ablations and experiments with a different dataset (LAION-400M) make the study more self-contained.
Many evaluations report error bars wherever appropriate.

5. **Excellent clarity in writing and presentation:**
The motivation for this study is precisely stated I the abstract and introduction.
The coverage of related work is broad and comprehensive.
All technical details for empirical analysis are well-stated and easy to follow.
The main paper and supplementary material have adequate implementation details to aid reproducibility.


**Weaknesses:**

I have some questions and suggestions that could make the study more comprehensive.
Have the authors considered the following experiments in their study?

1. **The exact autoregressive language model used in CLIP paper:**
CLIP paper uses a different autoregressive model, and this particular model is shown to converge slowly.
Have the authors tried this exact architecture?
Instead of a transformer decoder with cross-attention to image features (e.g. like VirTex),
CLIP's autoregressive baseline has a SimVLM-style design wherein image features are pooled into 2x2 grid and passed to the text model as the first four tokens.
The model follows the transformer encoder design and predicts caption tokens autoregressively.

2. **Backward captioning or masked language modeling?**
Have the authors considered auxiliary objectives used by prior works, such as backward captioning (VirTex) or masked language modeling (ICMLM)?
These objectives can amortize the cost of forward pass through the image encoder and provide denser gradients to the image encoder.

3. **[Related to above] multiple parallel decoders:**
The above suggestion can be extended to enable the use of multiple lightweight text decoders with multiple auxiliary objectives.
One can design each decoder head with reduced capacity to make all models have comparable sizes.
Training with such multiple objectives should speed up convergence.

4. **Evaluation on dense prediction tasks?**
If the goal of this study is to learn high-quality image encoders, then I suggest the authors may include additional evaluations
with dense prediction tasks like object detection and segmentation. These tasks are ubiquitous in vision and quite challenging.
I suggest that authors could train a ViTDet-style model with a frozen/fine-tunable image encoder from Cap/CapPa training.


**Questions:**

Minor suggestion:

- This paper references CLIP in many places in the text. However, it is sometimes awkward to read as "[40] showed..." or "released by [40] ...".
  Ultimately, it is personal preference, but I may recommend the authors use `citet` format like "Radford et al. [40] showed that ..."
  for a better reading experience if they do not have a preference.
- `Line 84`: GeLU -> GELU. ReLU ("Re" = "Rectified") and GELU ("GE" -> "Gaussian Error") :-)
- What is the "(ok:as)" in section 4.2 title? Seems like a latex macro :-)

**Limitations:**

The authors have included a reasonable discussion about the limitations of this study (and their trained captioning models)
in the final section of the paper.

---

> ### Author Rebuttal · Authors · 2023-08-09
>
> Thank you for suggesting additional experiments, many of which we were able to address.
> \
> \
> **Performance of the exact autoregressive language model used in CLIP paper**
>
> We trained this exact model (with a ResNet-50 encoder as in the CLIP paper) for 900M examples seen and found that it performs somewhat worse than the same encoder model together with the transformer decoder architecture. However, we believe that exploring alternative, potentially simpler decoder architectures at scale is an interesting direction for future work.
>
> Comparing prefix decoder with baselines (10-shot linear eval. and scoring)
> | model                         |   ImageNet |   CIFAR100 |   Pets |   Cars |   INet zs. |
> |:------------------------------|-----------:|-----------:|-------:|-------:|-----------:|
> | CLIP* (8k) R50                |       39.8 |       33.5 |   49.2 |   60.9 |       43.6 |
> | Cap R50 (transformer decoder) |       37.8 |       33.3 |   48.6 |   52.4 |       28.5 |
> | Cap R50 (prefix decoder)      |       36.8 |       30.4 |   41.5 |   45.4 |       27.6 |
>
> \
> **Cap/CapPa with backward captioning or masked language modeling**
>
> We explored masked language modeling in the context of parallel prediction by masking only a fraction of the tokens, but only observed improvements over pure autoregressive modeling when masking all tokens (see Sec. 4.4, Parallel prediction).
> As for backward captioning, we trained Cap while randomly reversing the caption with probability 0.5 (with the 900M examples schedule). This ensures that model capacity and pretraining compute remain unchanged. We do not observe improved performance (see below). While VirTex ablates backwards captioning and shows improvements, they use a separate decoder, so the ablated model has fewer parameters and FLOPs (here we control for both factors).
>
> Forward vs forward + backward captioning (ViT-B/16 encoder; 10-shot linear eval.)
> | model           |   ImageNet |   CIFAR100 |   Pets |   Cars |
> |:----------------|-----------:|-----------:|-------:|-------:|
> | Cap             |       49.7 |       56.0 |   72.6 |   74.7 |
> | Cap (fwd + bwd) |       49.2 |       56.1 |   71.7 |   73.0 |
>
> \
> **Using multiple parallel decoders**
>
> To address this point, we trained a CapPa variant with two parallel decoders, one for autoregressive prediction and another one for parallel prediction, each with 3 layers instead of 6. This model matches the pretraining compute of the default Cap/CapPa models with 6 decoder layers. While this model performs better than Cap with 3 decoder layers in linear 10-shot eval, it does not clearly outperform Cap with 6 decoder layers and performs worse than CapPa on 3 out of 4 eval sets.
>
> Comparing separate decoders with baselines (for ViT-B/16 encoder; 10-shot linear eval.)
> | model                  |   ImageNet |   CIFAR100 |   Pets |   Cars |
> |:-----------------------|-----------:|-----------:|-------:|-------:|
> | Cap (3 dec. layers)    |       48.7 |       53.7 |   73.5 |   73.7 |
> | Cap                    |       49.7 |       56.0 |   72.6 |   74.7 |
> | CapPa w/ sep. decoders |       49.5 |       54.9 |   75.8 |   79.0 |
> | CapPa                  |       50.4 |       57.4 |   76.2 |   78.5 |
>
> \
> **Evaluation on dense prediction tasks**
>
> CLIP and image/text pretrained models more generally seem less popular than supervised/self-supervised vision encoders for such supervised dense prediction tasks. However, vision-language models have become popular recently for open vocabulary semantic segmentation (see e.g. [a, b, c, d]) and the field is evolving rapidly. We believe it might be interesting to explore this direction with captioning models, but we leave this for future work as it would require a substantial extension of the scope.
>
> - [a] Ding et al., Decoupling zero-shot semantic segmentation. CVPR 2022
> - [b] Ma et al., Open-vocabulary Semantic Segmentation with Frozen Vision-Language Models. BMVC 2022
> - [c] Liang et al., Open-vocabulary semantic segmentation with mask-adapted CLIP. CVPR 2023.
> - [d] Mukhoti et al., Open Vocabulary Semantic Segmentation with Patch Aligned Contrastive Learning. CVPR 2023
>
>
> \
> **Minor issues**
>
> Thank you for raising these, we will address them in the next revision of the paper.

---

> > ### Comment · Reviewer_J9Jb · 2023-08-21
> > **Thank you for a thoughtful rebuttal**
> >
> > I thank the authors for accepting my suggestions and spending effort in running additional experiments! The results in the rebuttal overall look promising, and I encourage the authors to report them in the supplementary material. Please find specific responses below:
> >
> > **Performance of the exact autoregressive language model used in CLIP paper**
> >
> > The authors found this model to perform worse than their `Cap` model. I view this as a positive result — authors state that the observation of Radford et al., 2021 (the captioning models converge slower) mostly holds for a specific configuration they experimented with (ResNet-50 + 12-layer transformer) but disappears when scaling to ViTs and such. The exact autoregressive architecture used by CLIP is one more factor that contributed to their observation which the authors claim to (somewhat) refute in this paper.
> >
> > **Cap/CapPa with backward captioning or masked language modeling**
> >
> > I believe this experiment could be better designed — the authors use one set of transformer weights to process forward and reversed captions. Reversed (English) captions are like a different language that is composed of English words, but follow completely opposite sentence structure and grammar rules (e.g. `subject-verb-object` becomes `object-verb-subject`). The model is modeling two essentially different languages that use the same words, which intuitively sounds like a very difficult problem — one may argue it's more difficult than learning a multilingual language model with English and other known languages (with sensible grammar).
> >
> > > > While VirTex ablates backwards captioning and shows improvements, they use a separate decoder, so the ablated model has fewer parameters and FLOPs (here we control for both factors).
> >
> > I appreciate the authors' thoughtfulness in controlling for params/FLOPs. Indeed (Desai & Johnson) should have controlled for this by using two transformers intact but performing forward captioning with both. I think the updated experiment is not necessary — the central message of the paper still holds without it. I thank the authors for running this experiment!
> >
> > **Using multiple parallel decoders:** Thank you for running this experiment, it shows that downstream performance mostly depends on the params/FLOPs of the decoders. Performance is less sensitive to how these parameters are divided across multiple decoders for auxiliary language modeling tasks... perhaps to some extent, I bet having CapPa models with six decoders of one layer each may get weak (??)
> >
> > **Evaluation on dense prediction tasks:** The authors' argument is persuasive — while I believe that this paper could be enriched by these evaluations, I note that they are not crucial to back the main claims presented in the paper. I would discount this concern in my final assessment.
> >
> > ---------
> >
> > **Summary:** I recommended acceptance before the rebuttal. I continue to recommend acceptance after the authors' rebuttal. The paper presents a topic that would be of broad interest to the NeurIPS audience and presents it with sound experiments and high-quality presentation. While all reviewers unanimously agree to accept, I am happy to defend this paper for acceptance if needed. Congratulations to the authors!

---

### Author Rebuttal · Authors · 2023-08-09

We thank all the reviewers for carefully reading our paper. We appreciate their thoughtful comments and the overall very favorable assessment. In particular, we liked the many suggestions for additional experiments, some of which we were able to run. We believe these experiments will make the paper stronger.

Please find the detailed responses to each review below.

---

### Decision · Program_Chairs · 2023-09-21

**Decision:**

Accept (oral)

**Comment:**

The work received unanimous positive reviews. The work presented a comparison of contrastive and generative approaches in the context of visual representation learning from image-text pairs and demonstrate that when training data, compute, and model capacity are matched, generative approaches are competitive. Further, results on ARO benchmark provide evidence to suggest that generative approaches might have an advantage when representing relations, attributes, and order.

AC agrees with the reviewers and finds the work relevant to the vision-language community. Given the enthusiasm from reviewers and high-quality experiments, the AC is pleased to recommend this work for a **Spotlight**!

The AC also encourages the authors to continue investigating the advantages of generative approaches over contrastive approaches beyond the single ARO benchmark used in this work.